# Cancer stem cell regulated phenotypic plasticity protects metastasized cancer cells from ferroptosis

Mingming Wu [1,2,10], Xiao Zhang [1,2,10], Weijie Zhang[1,2], Yi Shiou Chiou [3,4,5], Wenchang Qian[1,2], Xiangtian Liu[1,2], Min Zhang[1,2], Hong Yan[6], Shilan Li[6], Tao Li[7], Xinghua Han[1], Pengxu Qian [8], Suling Liu [9], Yueyin Pan [1], Peter E. Lobie [3,5✉] & Tao Zhu [1,2✉]

Cancer cells display phenotypic equilibrium between the stem-like and differentiated states during neoplastic homeostasis. The functional and mechanistic implications of this sub-population plasticity remain largely unknown. Herein, it is demonstrated that the breast cancer stem cell (BCSC) secretome autonomously compresses the stem cell population. Co-implantation with BCSCs decreases the tumor-initiating capacity yet increases metastasis of accompanying cancer cells, wherein DKK1 is identified as a pivotal factor secreted by BCSCs for such functions. DKK1-promotes differentiation is indispensable for disseminated tumor cell metastatic outgrowth. In contrast, DKK1 inhibitors substantially relieve the metastatic burden by restraining metastatic cells in the dormant state. DKK1 increases the expression of SLC7A11 to protect metastasizing cancer cells from lipid peroxidation and ferroptosis. Combined treatment with a ferroptosis inducer and a DKK1 inhibitor exhibits synergistic effects in diminishing metastasis. Hence, this study deciphers the contribution of CSC-regulated phenotypic plasticity in metastatic colonization and provides therapeutic approaches to limit metastatic outgrowth.

[1] The First Affiliated Hospital of USTC, Division of Life Sciences and Medicine, University of Science and Technology of China, Hefei, Anhui, China. [2] The CAS Key Laboratory of Innate Immunity and Chronic Disease, Division of Life Sciences and Medicine, University of Science and Technology of China, Hefei, Anhui, China. [3] Tsinghua-Berkeley Shenzhen Institute and Institute of Biopharmaceutical and Health Engineering, Tsinghua Shenzhen International Graduate School, Shenzhen, China. [4] Master Degree Program in Toxicology, College of Pharmacy, Kaohsiung Medical University, Kaohsiung, Taiwan. [5] Shenzhen Bay Laboratory, Shenzhen, China. [6] Department of Pathology, The First Affiliated Hospital of USTC, Division of Life Sciences and Medicine, University of Science and Technology of China, Hefei, Anhui, China. [7] Department of Clinical Laboratory, The First Affiliated Hospital of Anhui Medical University, Hefei, Anhui, China. [8] Center for Stem Cell and Regenerative Medicine, Department of Basic Medical Sciences and Institute of Hematology, The First Affiliated Hospital, Zhejiang University School of Medicine, Hangzhou, Zhejiang, China. [9] Fudan University Shanghai Cancer Center & Institutes of Biomedical Sciences, Shanghai Medical College, Key Laboratory of Breast Cancer in Shanghai, Innovation Center for Cell Signaling Network, Cancer Institute, Fudan University, Shanghai, China. [10] These authors contributed equally: Mingming Wu, Xiao Zhang. ✉email: pelobie@sz.tsinghua.edu.cn; zhut@ustc.edu.cn

Breast cancer encompasses hierarchies with a self-renewing, highly metastatic and therapeutic resistant subpopulation, termed breast cancer stem cells (BCSCs), at the apex[1]. Sharing similar surface markers with stem cells and exhibiting growth as spheroids in serum-free suspension culture, BCSCs have been reported to drive cancer progression and relapse[2,3]. Upon homeostatic culture, cancer cells display stable stem-like and differentiated bulk populations during propagation. The phenotypic equilibrium of these tumor cell subpopulations may be described by a quantitative Makov model[4]. Either in culture or in vivo, relatively enriched BCSCs rapidly reestablish the composition of parental cells, which suggests the propensity of BCSCs to differentiate in such conditions[4,5]. The equilibrium between renewal and differentiation is presumably orchestrated by both intrinsic and extrinsic determinants. For example, in breast cancer, IL6 is secreted by differentiated cancer cells to support CSC survival and self-renewal[5]. Differentiated glioblastoma cells have also been reported to promote glioma stem cell self-renewal through BDNF-NTRK2-VGF signaling[6]. Therefore, loss of the supportive niches for CSCs, which are generated by differentiated cancer cells, is presumed to induce rapid differentiation of enriched CSCs. However, the autonomous mechanisms by which CSCs regulate the phenotypic plasticity of cell subpopulations remain largely unknown.

Metastasis is responsible for 90% of cancer-related deaths[7]. Intricately associated with CSC characteristics, the metastatic process is initiated by dedifferentiation coupled with acquisition of invasiveness in the primary tumor, which enhances cancer cell motility and dissemination[8,9]. Upon arrival at distant sites, disseminated cancer cells undergo redifferentiation during the process of post-extravasation metastatic colonization[10,11]. The dissemination of cancer cells seems to be an early and efficient step[12], followed by the survival and proliferation of disseminated cancer cells as a prerequisite for the establishment of macrometastases[11,13]. Strategies to limit or prevent metastatic outgrowth of disseminated tumor cells (DTCs) have increasingly gained attention as an approach to potentially reduce metastatic mortality. The DTCs exhibit marked similarities with CSCs and remain largely dormant after arriving at distant organs[14,15]. Reactivation from dormancy is required for DTCs to generate overt metastases. Metastatic cells from low-metastatic burden tissues exhibit a conserved stem cell signature, whereas high-metastatic burden tissue metastatic cells expressed higher levels of genes associated with differentiation[16]. As the invasive/de-differentiated cancer cells are predominantly growth-arrested, redifferentiation at distant sites may be required to exit dormancy for further metastatic outgrowth[17]. However, the mechanism by which DTCs undergo differentiation, exit quiescence and subsequent metastatic colonization remain poorly characterized.

Ferroptosis, a non-apoptotic cell death process driven by aberrant metabolism and iron-dependent lipid peroxidation, has been implicated in a variety of pathologies including cancer. Increasing evidence has suggested that several oncogenic signaling pathways render cancer cells extremely susceptible to ferroptosis[18]. Previous studies also suggested that highly aggressive mesenchymal-like cancer cells are vulnerable to ferroptosis[19,20]. Constitutionally associated with mesenchymal cancer cells, the mesenchymal-like CSC population has also been identified to be highly susceptible to ferroptosis-induced cell death[21,22]. Interestingly, salinomycin-mediated accumulation of iron in lysosomes selectively eliminates CSCs by inducing ferroptosis[23]. Therefore, the differentiation of CSCs is presumably to protect cancer from ferroptosis. Moreover, cancer cells that have metastasized to lung experience higher oxidative stress and tend to be eliminated more frequently by oxidative stress-induced ferroptosis compared to that in primary mammary or subcutaneous cancers[24,25]. The differentiation of DTCs is presumably maintained to protect metastases from ferroptosis.

Intrigued by the CSC initiated autonomous regulation of cell subpopulations, co-culture systems in vitro and co-implantation systems in vivo were designed to characterize the functional and mechanistic implications of this phenomenon. By screening the function of the BCSC-derived secretome in the regulation of cancer cell phenotypic plasticity, DKK1 was identified as a pivotal molecule that autonomously diminishes the CSC population and subsequently promotes breast cancer metastatic colonization. DKK1 also significantly reduced ferroptosis-specific lipid peroxidation and induced a ferroptosis-resistant cell state. Thus, blockade of DKK1 combined with induction of ferroptosis may be a potential therapeutic strategy to limit metastatic disease.

## Results

**BCSC secretome compresses the stem cell pool**. To study BCSC-regulated phenotypic plasticity of different cellular subpopulations, the ALDEFLUOR assay was used to separate the BCSCs and differentiated cancer cell populations[26]. In monolayer culture, enriched ALDH+ BCSCs rapidly gave rise to ALDH- cells, leading to a rapid and potent decrease in the percentage of ALDH+ cells in the cohort, while ALDH− differentiated cancer cells more slowly generated ALDH+ cells over time (Fig. 1a). To illuminate the mechanism underlying the rapid differentiation of enriched BCSCs, we co-cultured RFP-labeled T47D cells with FACS-sorted unlabeled ALDH−, ALDH+ or parental cells in monolayer culture, respectively. The proportion of ALDH+ cells in the RFP+ cell population was decreased by 70% in the presence of untagged ALDH+ T47D cells compared to co-culture with the same number of untagged parental cells (Fig. 1b and Supplementary Fig. 1a), suggesting the negative regulation of the stem cell pool by enriched BCSCs. In comparison, the proportion of ALDH+ cells in the RFP+ cell population was appreciably increased in the presence of untagged ALDH− T47D cells compared to the co-culture with the same number of untagged parental cells (Fig. 1b and Supplementary Fig. 1a). To determine whether such an effect is cell contact-dependent, T47D cells were cultured with the conditioned medium (CM) derived from ALDH−, ALDH+ or parental T47D cells, respectively. As shown in Fig. 1c and Supplementary Fig. 1b, CM derived from ALDH+ cells resulted in a significant decrease, while CM from ALDH− cells slightly increased, the percentage of ALDH+ cells, suggesting that the observed effects are not cell contact-dependent. Mammosphere-enriched BCSCs were further utilized and verified exhibiting significant enrichment of CD44+CD24−/low cells with elevated expression of multiple stemness markers (Supplementary Fig. 1c, d). Further, either mammosphere-enriched BCSCs or parental cells were seeded in the upper chambers, whereas the parental cells were seeded in the lower compartment of the co-culture system (Fig. 1d). The percentage of the ALDH+ population or mammosphere-formation efficiency of the cells in the lower compartment was determined 48 h later. Consistently, the percentage of the ALDH+ population and mammosphere-formation capacity were significantly decreased when MCF-7 or T47D cells were co-cultured with mammosphere-enriched BCSCs (Fig. 1e, f and Supplementary Fig. 1e). The same effects were observed in the cells cultured with CM derived from mammosphere-enriched BCSCs or parental cells (Fig. 1g, h and Supplementary Fig. 1f). Additionally, the CD44+CD24−/low BCSC population was also decreased in MCF-7 or T47D cells cultured with BCSC CM compared to that of control CM (Supplementary Fig. 1g). Similar results were also observed using HER2+ BT474 cells and triple-negative (TNBC) SUM159 cells (Supplementary Fig. 1h, i), suggesting a generic role of the BCSC-derived secretome in regulating the CSC pool in breast cancer cells. Cell proliferation was nevertheless not affected in the presence of

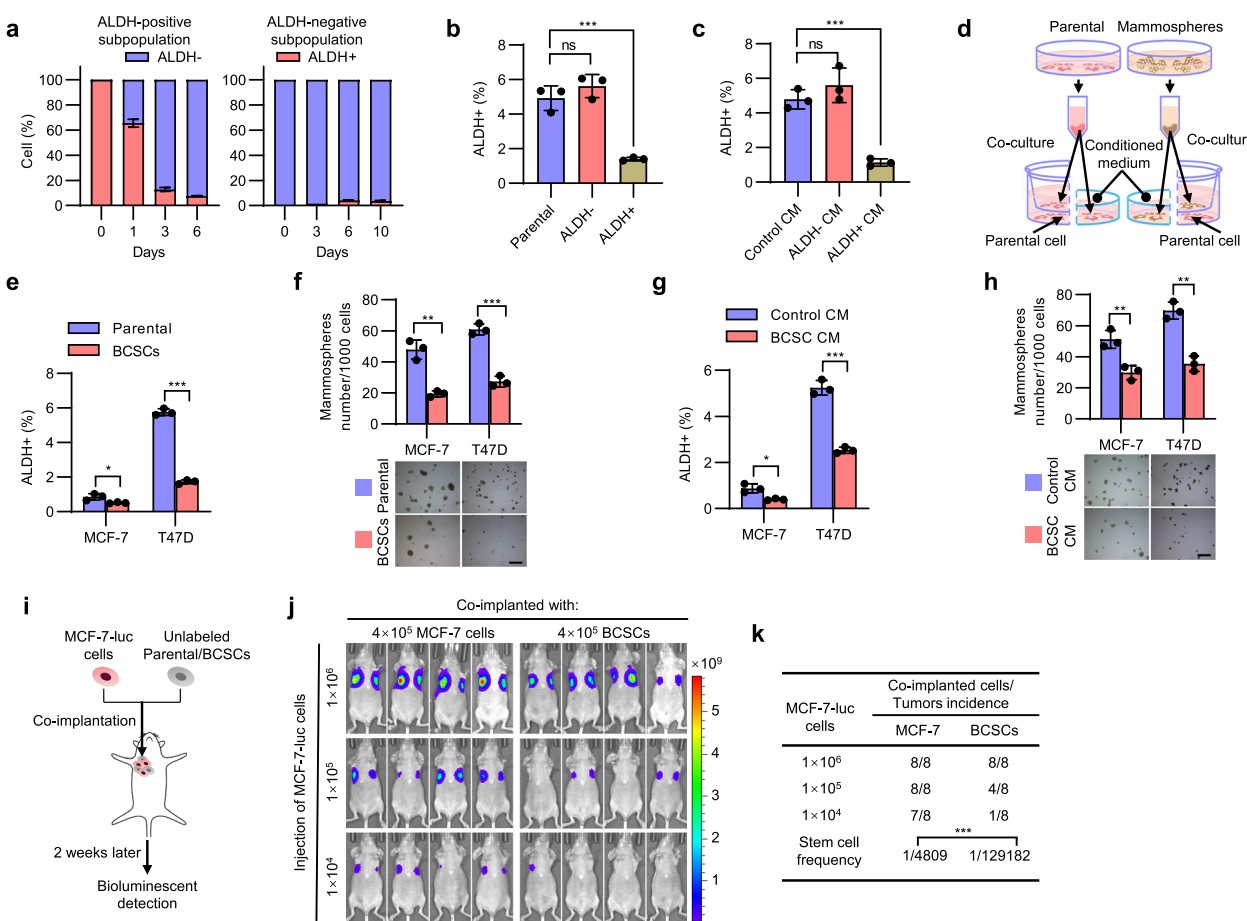

**Fig. 1 BCSC secretome compresses the stem cell pool size. a** FACS analysis of the proportion of ALDH+ or ALDH− cells after the growth of FACS-sorted ALDH+ or ALDH− T47D cells. **b** FACS analysis of the proportion of ALDH+ BCSCs in the RFP-labeled T47D cells co-cultured with the unlabeled ALDH−, ALDH+ or parental T47D cells. **c** FACS analysis of the proportion of ALDH+ BCSCs in the T47D cells cultured with the CM derived from ALDH+, ALDH−, or parental cells. **d** The schematic of conditioned medium and transwell co-culture system. **e, f** MCF-7 or T47D cells were co-cultured with mammosphere-enriched BCSCs or parental cells for 48 h, and the stemness properties were subsequently analyzed by ALDEFLUOR assay (**e**) or mammosphere-formation assay (**f**). Scale bars: 500 μm. **g, h** MCF-7 or T47D cells were cultured with the respective CM derived from mammosphere-enriched BCSCs or parental cells for 48 h, and the stemness properties were subsequently analyzed by ALDEFLUOR assay (**g**) or mammosphere-formation assay (**h**). Scale bars: 500 μm. **i** The schematic of co-implantation model. **j, k** A series of limiting diluted MCF-7-luc cells were co-implanted with unlabeled 4 × 10⁵ mammosphere-enriched BCSCs or parental cells into host mice. Bioluminescent imaging (BLI) was performed on tumors generated by MCF-7-luc cells (**j**), the CSC frequency was calculated using ELDA software (**k**). Results are shown as mean ± S.D. *P < 0.05; **P < 0.01; ***P < 0.001; ns not significant (One-way ANOVA followed by Tukey's multiple comparison test in (**b**, **c**) others unpaired two-tailed Student's *t* test). Source data are provided as a Source Data file.

BCSC-derived CM (Supplementary Fig. 1j). To determine the tumor-initiating capacity in vivo, a series of limiting diluted luciferase-labeled MCF-7 cells were co-implanted with unlabeled 4 × 10⁵ mammosphere-enriched BCSCs or parental MCF-7 cells into the second mammary fat pads of female nude mice (Fig. 1i). The tumor-initiating capacity of MCF-7-luc cells in mice co-injected with BCSCs was significantly reduced compared with the control mice as determined by bioluminescent imaging 2 weeks after implantation (Fig. 1j, k). These results suggest that BCSCs compress the CSC pool and subsequent tumor-initiating capacity.

**BCSC-secreted DKK1 shrinks the stem cell pool.** To determine the mechanism by which BCSCs autonomously compress their pool size, the influence of BCSC-derived CM on three canonical signaling pathways of stem cells, including WNT/β-CATENIN, NOTCH and HEDGEHOG, were examined[27]. NICD and GLI2, which indicate the activity of NOTCH and HEDGEHOG

pathways respectively, did not appear to be negatively affected by the CM from BCSCs (Supplementary Fig. 2a), whereas β-CATENIN levels, a marker for canonical WNT signaling, were remarkably downregulated in both MCF-7 and T47D cells exposed to BCSC CM (Fig. 2a). Furthermore, the luciferase activity of a β-CATENIN reporter decreased by 70% in BCSC-derived CM (Fig. 2b). Consistently, the expression levels of WNT/β-CATENIN target genes AXIN2, LGR5 and c-MYC were decreased by BCSC-derived CM (Supplementary Fig. 2b). This observation indicated that the BCSC secretome attenuated WNT/β-CATENIN signaling in parental cells. In silico analysis was further performed to compare the expression levels of 8 known WNT/β-CATENIN antagonists in ALDH+ and ALDH− cells using the published datasets[28]. *DKK1* was the most abundant and significantly upregulated gene in the ALDH+ cells from both MCF-7 and T47D cells (Fig. 2c). The endogenous levels of DKK1 in mammosphere-enriched BCSCs were significantly elevated compared to the parental cells (Fig. 2d). Furthermore, the

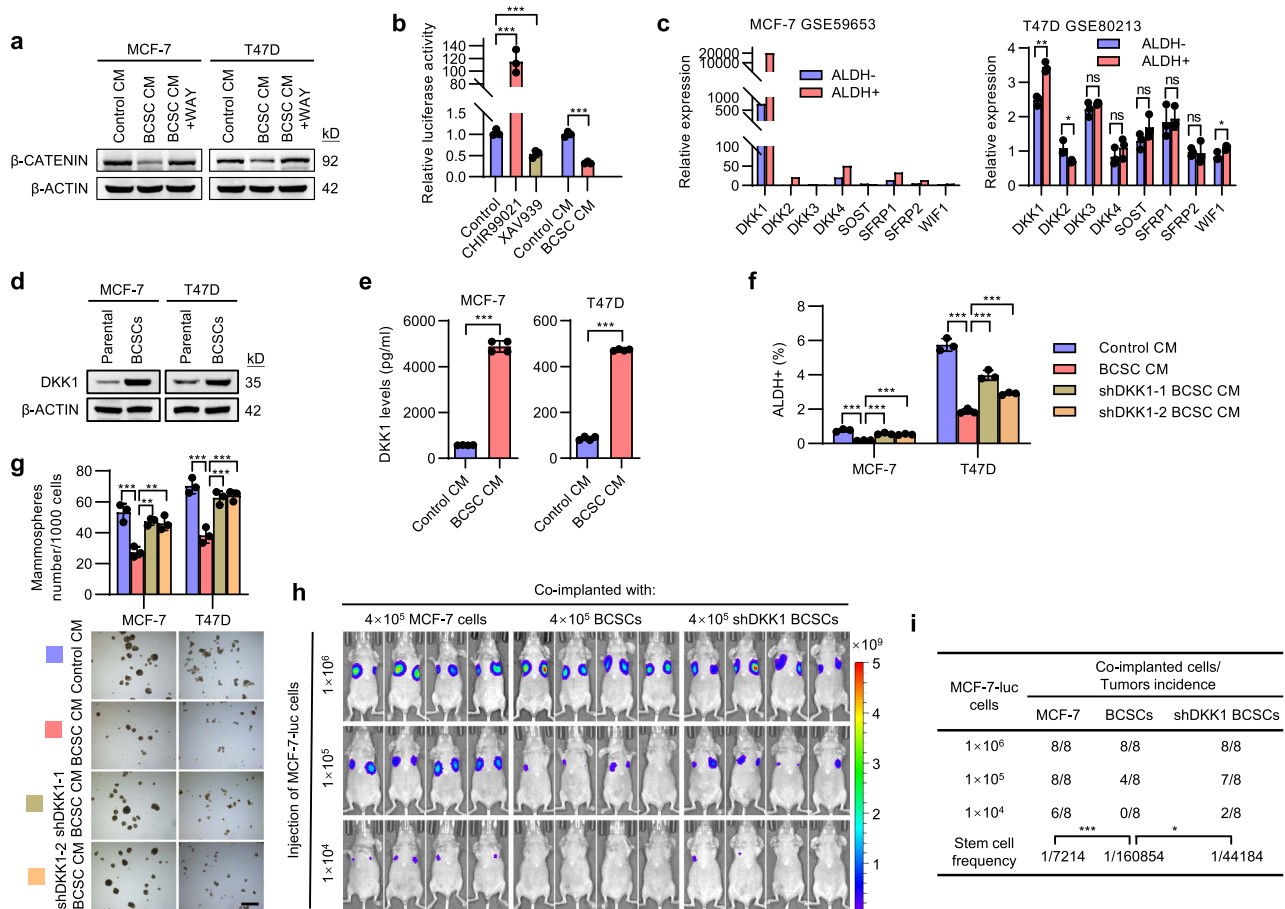

**Fig. 2 BCSC-secreted DKK1 shrinks the stem cell pool. a** Immunoblot assessment of β-CATENIN protein levels. Cells were cultured with the CM obtained from respective parental cells or mammosphere-enriched BCSCs and treated with DKK1 Inhibitor WAY262611 (1 μM) or vehicle. **b** Luciferase activities of the β-CATENIN reporter 8 × TOP Flash in MCF-7 cells cultured with the respective CM obtained from parental cells or mammosphere-enriched BCSCs, the β-CATENIN agonist CHIR99021 (1 μM) or inhibitor XAV939 (1 μM) served as the positive or negative control, respectively. **c** The gene expression profiles of pertinent WNT/β-CATENIN antagonists in matched pairs of ALDH+ and ALDH− MCF-7 or T47D cells. The data were downloaded from GEO GSE59653 or GSE80213, respectively. **d** Immunoblot assessment of DKK1 in the matched pairs of mammosphere-enriched BCSCs and parental cells. **e** ELISA quantification of secreted DKK1 levels in CM from paired mammosphere-enriched BCSCs or parental cells. **f, g** MCF-7 or T47D cells were cultured with the CM derived from respective parental cells or shCONT- or shDKK1-derived mammospheres for 48 h, the stemness properties were subsequently analyzed by ALDEFLUOR assay (**f**) or mammosphere-formation assay (**g**). Scale bars: 500 μm. **h, i** A series of limiting diluted luciferase-labeled MCF-7 cells were co-implanted with 4 × 10⁵ unlabeled parental cells or shCONT- or shDKK1-derived mammosphere-enriched BCSCs into the second fat pad of host mice. BLI was performed on the tumor burden generated by MCF-7-luc cells (**h**), the CSC frequency was calculated using ELDA software (**i**). Results are shown as mean ± S.D. *P < 0.05; **P < 0.01; ***P < 0.001; ns not significant (Unpaired two-tailed Student's *t* test in (**c**, **e**), others one-way ANOVA followed by Tukey's multiple comparison test). Source data are provided as a Source Data file.

secreted levels of DKK1 were also significantly upregulated in CM from either mammospheres or ALDH+ cells compared to that from control cells (Fig. 2e and Supplementary Fig. 2c).

DKK1 was reported to inhibit WNT signaling by binding to WNT receptor LRP5/6 directly[29]. Consistently, supplementation of recombinant DKK1 in the medium reduced β-CATENIN levels, the percentage of ALDH+ cells as well as mammosphere-formation capacity in both MCF-7 and T47D cells (Supplementary Fig. 2d–f). Furthermore, MCF-7 and T47D cells were cultured in the presence of CM from BCSCs stably transfected with shCONT or shDKK1 (Supplementary Fig. 2g). Whereas the CM from shCONT BCSCs significantly decreased the ALDH+ cell population and mammosphere-formation capacity, DKK1 depletion in BCSCs inhibited these effects (Fig. 2f, g and Supplementary Fig. 2h). Application of a small molecule DKK1 antagonist, WAY262611[30], largely rescued the decreased β-CATENIN level, the percentage of ALDH+ cells and mammosphere-formation capacity afforded by BCSC-derived

CM in MCF-7 and T47D cells (Fig. 2a and Supplementary Fig. 2i, j). Consistently, co-injection of MCF-7-luc cells with unlabeled shDKK1 BCSCs reversed decreased tumor-initiating capacity afforded by co-injection with control BCSCs (Fig. 2h, i). Hence, these data suggest that BCSC-derived DKK1 shrunk the CSC pool and reduced the tumor-initiating capacity in breast cancer cells.

**BCSC-secreted DKK1 enhances metastatic colonization.** Prevailing theories suggest that metastases are predominantly initiated by rare cancer cells with unique CSC properties[31,32]. However, the proportion of CSCs in metastatic sites is enriched at the beginning of the metastasis and decreases rapidly during the advanced metastatic stages[11]. We therefore determined whether the autonomous compression of enriched BCSCs, which was regulated by BCSC-derived DKK1, exerted a role in cancer metastasis by using triple-negative breast cancer (TNBC) cell lines. The enriched unlabeled SUM159 BCSCs were co-injected

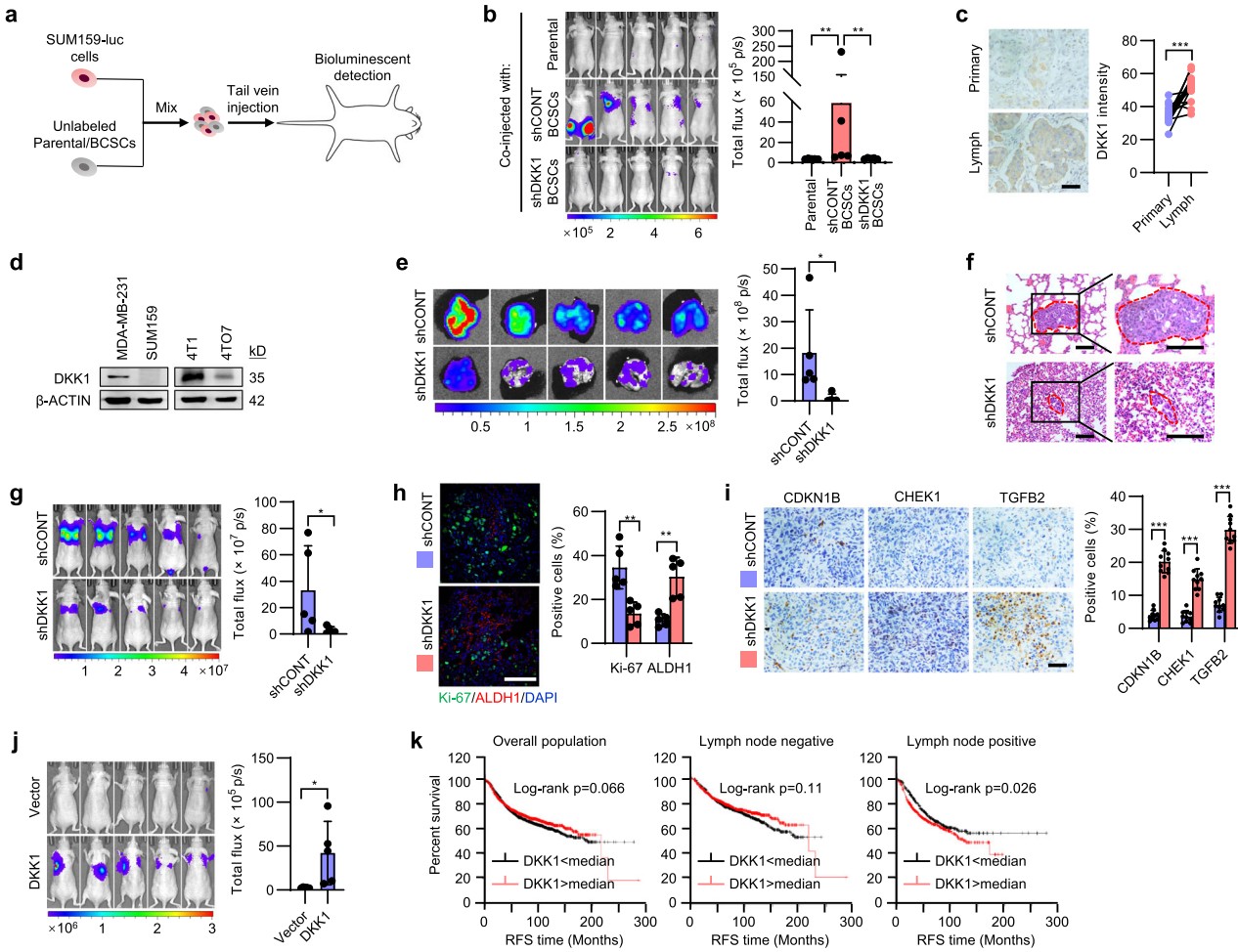

**Fig. 3 BCSC-secreted DKK1 enhances metastatic colonization. a** The schematic of co-implantation model. The 1 × 10⁶ luciferase-labeled SUM159-luc cells were intravenously co-injected with 4 × 10⁵ unlabeled mammosphere-enriched shCONT BCSCs, shDKK1 BCSCs or parental cells into the host mice. BLI was performed on the metastatic burden of SUM159-luc cells. **b** BLI was performed on the metastatic burden of SUM159-luc cells. **c** Immunohistochemical (IHC) staining of DKK1 in 15 pairs of human primary breast cancer and lymph node metastases. Scale bar: 50 μm. **d** Immunoblot assessment of DKK1 protein levels. **e, f** BLI (**e**) or H&E staining (**f**) of the lungs from mice orthotopically implanted with 2 × 10⁶ MDA-MB-231-shCONT or -shDKK1 cells 4 weeks later. Scale bar: 100 μm. **g** BLI of the metastatic burden of mice intravenously injected with 1 × 10⁶ MDA-MB-231-shCONT or -shDKK1 cells. **h, i** Immunofluorescent (IF) staining of Ki-67 and ALDH1 (**h**) or IHC staining of CDKN1B, CHEK1 and TGFB2 (**i**) in lung sections from mice intravenously injected with MDA-MB-231-shCONT or -shDKK1 cells. Scale bar in (**h**): 100 μm. Scale bar in (**i**): 50 μm. **j** BLI of the metastatic burden of mice intravenously injected with 1 × 10⁶ SUM159-Vector or SUM159-DKK1 cells. **k** Kaplan–Meier plots of relapse-free survival (RFS) in the indicated population of breast cancer patients stratified according to tumor DKK1 expression. The data was analyzed in all molecular subtypes of breast cancer. Results are shown as mean ± S.D. *$P < 0.05$; **$P < 0.01$; ***$P < 0.001$; ns not significant (Paired two-tailed Student's $t$ test in (**c**) others unpaired two-tailed Student's $t$ test). Source data are provided as a Source Data file.

with luciferase-labeled parental cells into the tail vein of nude mice. The metastatic burden derived from the co-injected luciferase-labeled parental cells was determined by bioluminescent imaging (Fig. 3a). Strikingly, BCSCs were sufficient to convert the accompanying non-metastatic SUM159-luc cells to highly metastatic, while co-injection of DKK1-depleted BCSCs abrogated this effect (Fig. 3b and Supplementary Fig. 3a), indicating a critical role of the BCSC secretome in promoting cancer metastasis. Consistently, BCSCs derived from murine mammary carcinoma 4TO7 cells also afforded metastatic capacity to the accompanying inefficiently-metastatic 4TO7-luc cells (Supplementary Fig. 3b). Furthermore, SUM159-luc cells were cultured with CM derived from parental cells or BCSCs for 48 h and subsequently injected into the tail vein of nude mice and lung metastases were determined 30 days later. Although the result was not statistically significant, one out of five mice developed potent lung metastasis

derived from the non-metastatic SUM159 cells cultured with BCSC CM (Supplementary Fig. 3c). These findings suggest a potential role of BCSC-regulated phenotypic plasticity in the promotion of breast cancer metastatic progression.

To further determine the implications of DKK1-regulated differentiation in breast cancer metastasis, DKK1 expression in 15 paired invasive human breast cancer tissues was profiled. We observed that DKK1 expression was consistently elevated in lymph node metastases compared to their respective primary tumors (Fig. 3c). Furthermore, the highly metastatic MDA-MB-231 showed higher DKK1 levels compared to the non-metastatic SUM159 (Fig. 3d), although these two cell lines exhibited similar cell morphology and migratory/invasive capacity (Supplementary Fig. 3d, e). Similarly, in syngeneic murine mammary carcinoma cells, weakly-metastatic 4TO7 cells expressed much less DKK1 compared to highly metastatic 4T1 cells (Fig. 3d), although 4TO7

exhibited mesenchymal-like morphology and increased migrative/invasive capacity compared to epithelial-like 4T1 cells (Supplementary Fig. 3f–g). For further characterization, the endogenous DKK1 expression in MDA-MB-231 cells was depleted by shRNA and an increased CSC population was observed as a result compared to the control cells (Supplementary Fig. 3h, i). Consistent with in vitro cell migration/invasion assay data (Supplementary Fig. 3j), xenografts derived from orthotopically inoculated MDA-MB-231-shCONT or -shDKK1 cells exhibited a similar extent of local infiltration (Supplementary Fig. 3k). After 4 weeks, orthotopically injected MDA-MB-231-shCONT cells spontaneously formed pulmonary metastasis, in contrast, DKK1 depletion potently decreased the metastatic burden without significantly affecting the growth of the primary tumor (Fig. 3e and Supplementary Fig. 3l, m). Quantification of metastatic foci in lung sections suggested that the seeding efficiency of DTCs was not affected by DKK1 depletion (Supplementary Fig. 3n). Instead, the size of metastatic foci was potently diminished by DKK1 depletion (Fig. 3f), suggestive of an effect on post-dissemination outgrowth. To minimize the possible variation of primary tumor initiated distant metastasis, the tail vein metastatic model was used. Pulmonary seeding of tumor cells was shown not to be affected by DKK1 depletion as examined by bioluminescent imaging 4 h after cell injection (Supplementary Fig. 3o). Consistently, a much decreased metastatic burden was observed in the DKK1-depleted MDA-MB-231 cell group compared to the control group after 3 weeks (Fig. 3g and Supplementary Fig. 3p). Immunofluorescent co-staining of ALDH1 and Ki-67 in lung sections revealed that Ki-67 almost exclusively marked ALDH1-negative tumor cells, indicating the quiescent status of most BCSCs in the lung metastases (Fig. 3h). Interestingly, loss of DKK1 expression significantly increased the percentage of ALDH1+ BCSCs yet decreased the percentage of Ki-67+ proliferative cells in lung metastases (Fig. 3h). In addition, DKK1-depleted metastatic cells exhibited higher levels of stem cell markers, including KLF4, LIN28, NANOG and OCT3/4 as well as lower levels of cell cycle-associated genes such as CCND1 and CDK4 compared to the control (Supplementary Fig. 3q, r). DKK1 depletion did not alter the proportion of apoptotic cells in the metastatic sites (Supplementary Fig. 3s). A shift towards a more quiescent and dormant signature in DKK1-depleted metastatic cells, which expressed higher levels of CDKN1B, CHEK1 and TGFB2[16] was observed (Fig. 3i). Consistently, forced expression of DKK1 overtly converted the non-metastatic SUM159 cells to metastatic as observed in the tail vein metastatic model (Fig. 3j and Supplementary Fig. 3t).

Similarly, in an orthotopically implanted mouse mammary tumor model, DKK1 depletion in 4T1 cells abrogated pulmonary metastases without affecting the local infiltration of the primary tumors (Supplementary Fig. 4a–c). Again, in the tail vein injected metastatic model, DKK1 depletion in 4T1 cells did not affect pulmonary seeding efficiency compared to the control cells (Supplementary Fig. 4d), whereas a much decreased metastatic burden was observed in the DKK1-depleted group compared to the control group (Supplementary Fig. 4e). Consistently, forced expression of DKK1 in weakly-metastatic 4TO7 cells rendered it metastatic in the tail vein metastatic model (Supplementary Fig. 4f, g).

Examination of clinical data in breast cancer patients showed that although DKK1 expression was not significantly associated with survival outcome in the overall patient population, higher DKK1 expression was correlated with poorer survival outcome in lymph node-positive patients but not in lymph node-negative patients (Fig. 3k), hence supporting a functional impact of DKK1 in metastatic breast cancer. Furthermore, prognostic analysis of DKK1 expression in clinical samples from multiple cancers including gastric, lung, pancreatic, and head-neck squamous cell cancer (head-neck SCC) consistently revealed that patients with higher DKK1 expression exhibited worse overall survival outcomes (Supplementary Fig. 4h). Collectively, DKK1 promotes metastatic colonization of cancer cells without an impact on early cancer cell infiltration.

**Pharmacological inhibition of DKK1 abrogates metastatic progression.** The therapeutic potential of pharmacological DKK1 inhibition in breast cancer metastasis was further evaluated. DKK1 inhibitors, WAY262611 or Gallocyanine[30,33], remarkably reduced colony formation of MDA-MB-231 cells in 3D culture ex vivo (Supplementary Fig. 5a); concomitantly increased β-CATENIN levels were observed upon treatment with the inhibitors (Supplementary Fig. 5b). To determine if DKK1 inhibition could reduce metastatic burden, MDA-MB-231 cells were orthotopically injected into nude mice, and DKK1 inhibitors or vehicle (saline) were subsequently administered when the xenografts were visible on day 4. After a 4-week treatment, WAY262611 or Gallocyanine significantly diminished the pulmonary metastases derived from MDA-MB-231 cells (Fig. 4a), with the metastatic foci size also significantly reduced (Fig. 4b). Assessment of the body weight, hematological and blood biochemical indices of non-tumor bearing mice treated with WAY262611 or Gallocyanine indicated that only Gallocyanine had a slight influence on indicators of liver function (Supplementary Fig. 5c and Supplementary Table 1). Hematoxylin-eosin (H&E) staining indicated no significant liver pathology in Gallocyanine treated mice (Supplementary Fig. 5d). Thus, these inhibitors did not exert a significant influence on general animal health. The tail vein metastasis model was therefore further utilized. Twenty-four hours after tail vein injection, the mice were treated with WAY262611, Gallocyanine or saline as indicated. Whereas WAY262611 treatment resulted in a 50% reduction of pulmonary metastatic burden, Gallocyanine almost completely prevented the metastatic burden after a 3-week treatment (Fig. 4c and Supplementary Fig. 5e). Consistently, metastatic cells in the lung of treatment groups exhibited higher levels of stem cell markers but lower levels of cell cycle-associated genes (Supplementary Fig. 5f, g). WAY262611 or Gallocyanine also significantly decreased the percentage of Ki-67+ proliferative cells and increased the expression levels of cell dormancy-associated genes in the metastatic sites compared to the control group (Fig. 4d, e). Due to the higher efficacy of Gallocyanine, we further utilized Gallocyanine for determining the effect of DKK1 inhibition on the survival of mice with metastatic disease. In the vehicle-treated control group, all mice injected with $1 \times 10^6$ MDA-MB-231 cells via the tail vein died at around day 30 (Fig. 4f). In contrast, most Gallocyanine treated mice survived well beyond 50 days (Fig. 4f). Further, Gallocyanine also abrogated lung metastases derived from tail vein injected 4T1 cells and significantly extended the survival of host mice (Supplementary Fig. 5h, i). By using mouse mammary tumor virus-Polyoma virus middle T-antigen (MMTV-PyMT) transgenic mice, it was also shown that Gallocyanine abrogated lung metastases spontaneously derived from primary mammary tumors (Fig. 4g). MMTV-PyMT mice treated with Gallocyanine also exhibited significantly longer survival time compared to those treated with vehicle (Fig. 4h). To explore the therapeutic potential in a clinically-relevant context, the basal-like breast cancer PDX line, USTC-1, was orthotopically implanted in female NOD/SCID mice. Whereas mice in the control group exhibited abundant lung metastasis, no macro-metastases were visible in the WAY262611 or Gallocyanine treated groups (Fig. 4i, j). DKK1 has been reported to modulate NK-cell-mediated metastatic latency by

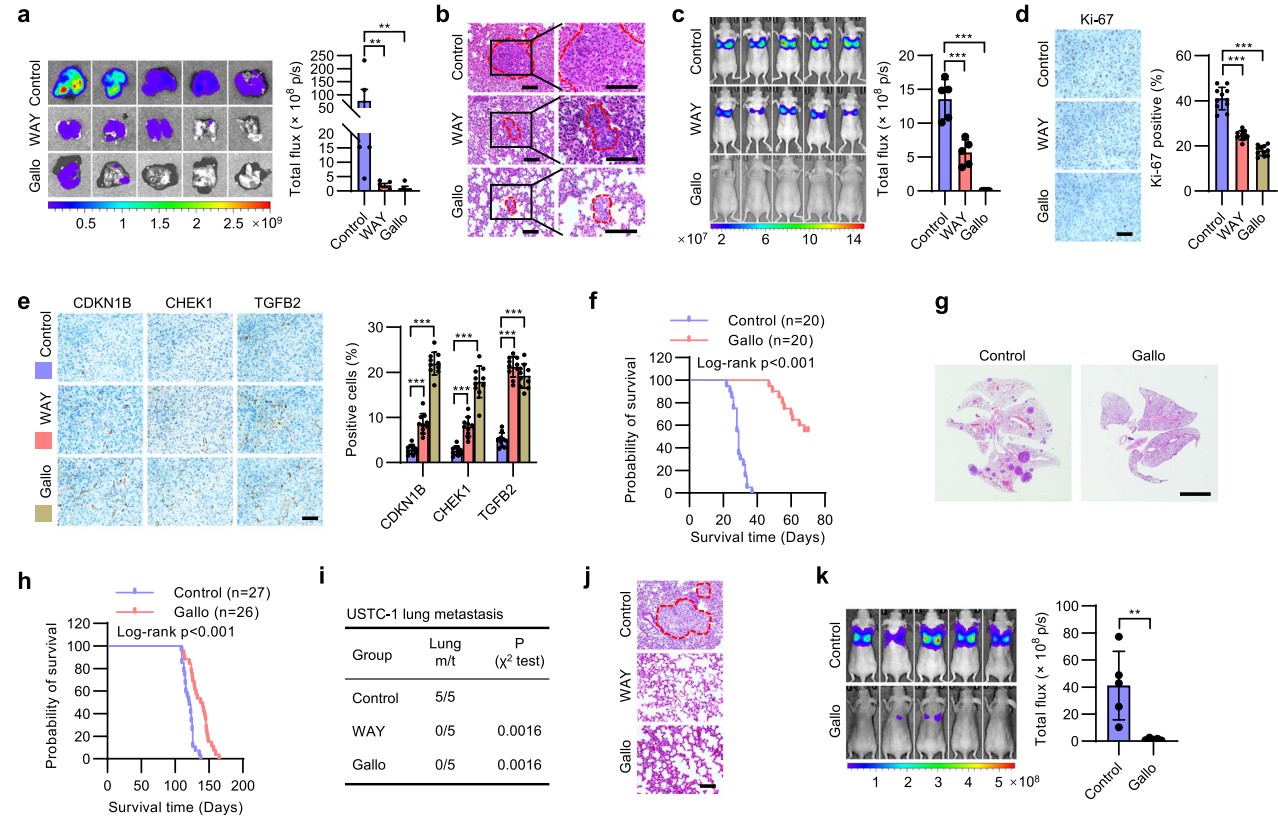

**Fig. 4 Targeting DKK1 ameliorates metastatic progression. a**, **b** BLI (**a**) and H&E staining (**b**) of the lungs from mice orthotopically implanted with $2 \times 10^6$ MDA-MB-231-luc cells. The mice were treated with vehicle, WAY262611 or Gallocyanine. Metastatic sites are circled in red. Scale bar: 100 μm. **c** BLI of the metastatic burden of mice intravenously injected with $1 \times 10^6$ MDA-MB-231-luc cells. The mice were treated with vehicle, WAY262611 or Gallocyanine. **d**, **e** IHC staining of Ki-67 (**d**) or CDKN1B, CHEK1 and TGFB2 (**e**) in lung sections from mice intravenously injected with MDA-MB-231-luc cells. The mice were treated with vehicle, WAY262611 or Gallocyanine. Scale bar: 50 μm. **f** Survival curve of mice intravenously injected with $1 \times 10^6$ MDA-MB-231-luc cells and treated with vehicle or Gallocyanine. **g** H&E staining of lung metastasis in MMTV-PyMT mice treated with vehicle or Gallocyanine. Scale bar: 5 mm. **h** Survival curve of MMTV-PyMT mice treated with vehicle or Gallocyanine. **i**, **j** Incidence (**i**) and H&E staining of lung metastasis (**j**) in mice orthotopically implanted with the PDX line USTC-1 and treated with vehicle, WAY262611 or Gallocyanine. Metastatic sites are circled in red. Scale bar: 100 μm. **k** BLI of the metastatic burden of NK-cell-depleted BALB/c nude mice intravenously injected with $5 \times 10^5$ MDA-MB-231-luc cells. The mice were treated with vehicle or Gallocyanine. Results are shown as mean ± S.D. *$P < 0.05$; **$P < 0.01$; ***$P < 0.001$; ns not significant (Unpaired two-tailed Student's $t$ test in (**a**, **c**, **k**) others one-way ANOVA followed by Tukey's multiple comparison). Source data are provided as a Source Data file.

suppression of NK-cell activating ligands[14]. To determine whether NK cells are involved in the anti-metastatic effects of DKK1 inhibition, we depleted BALB/c nude mice of NK cells by administration of polyclonal anti-asialo-GM1 antibody[14] and observed that Gallocyanine consistently abrogated lung metastases derived from tail vein injected MDA-MB-231 cells (Fig. 4k). A similar result was observed in severe immunodeficient NOD-Prkd$^{cem26Cd52}$Il2rg$^{em26Cd22}$/NjuCrl (NCG) mice (no NK cells) tail vein injected with 4T1 cells followed by treatment with Gallocyanine (Supplementary Fig. 5j), suggesting an NK-cell independent mechanism of DKK1-mediated metastatic outgrowth. Thus, pharmacological inhibition of DKK1 might be a potential strategy against metastatic disease.

**BCSC-secreted DKK1 protects cancer cells from ferroptosis**. It is fascinating that the BCSC secretome decreased primary tumor-initiating capacity yet increased distant metastasis. To further understand why the autonomous restraint of BCSCs could enhance metastatic colonization, we performed unbiased RNA sequencing in cells cultured with CM from BCSCs or parental cells. Pathway enrichment analysis suggested that metabolic

pathways, biosynthesis of amino acids and ferroptosis were among the most enriched pathways in cells cultured with BCSC CM (Fig. 5a). Cellular metabolism and amino acid biosynthesis have been reported to play crucial roles in ferroptosis[34]. The microenvironment of lung metastases exhibits higher ferroptotic stress compared to primary mammary cancers[25], resulting in altered dependence of ferroptosis. We therefore examined if the BCSC secretome-regulated decrease of the CSC pool modulated the sensitivity to ferroptosis. Erastin, a ferroptosis agonist, dramatically reduced the percentage of the ALDH+ population of MCF-7, MDA-MB-231, and 4T1 cells (Supplementary Fig. 6a), indicative that ferroptosis selectively eliminates BCSCs. To determine if the BCSC secretome resulted in reduced sensitivity to ferroptosis, MCF-7, MDA-MB-231, and 4T1 cells were cultured with CM derived from BCSCs or parental cells and subsequently treated with graded concentrations of Erastin for 48 h. Application of liproxstatin-1, a ferroptosis antagonist, partially reduced Erastin-induced cell death at high concentrations of Erastin (Fig. 5b). Although the non-specific toxicities of Erastin at high concentrations should also be noted, all cells cultured with BCSC CM developed persistent resistance to Erastin-induced ferroptosis at both high and low concentrations of Erastin,

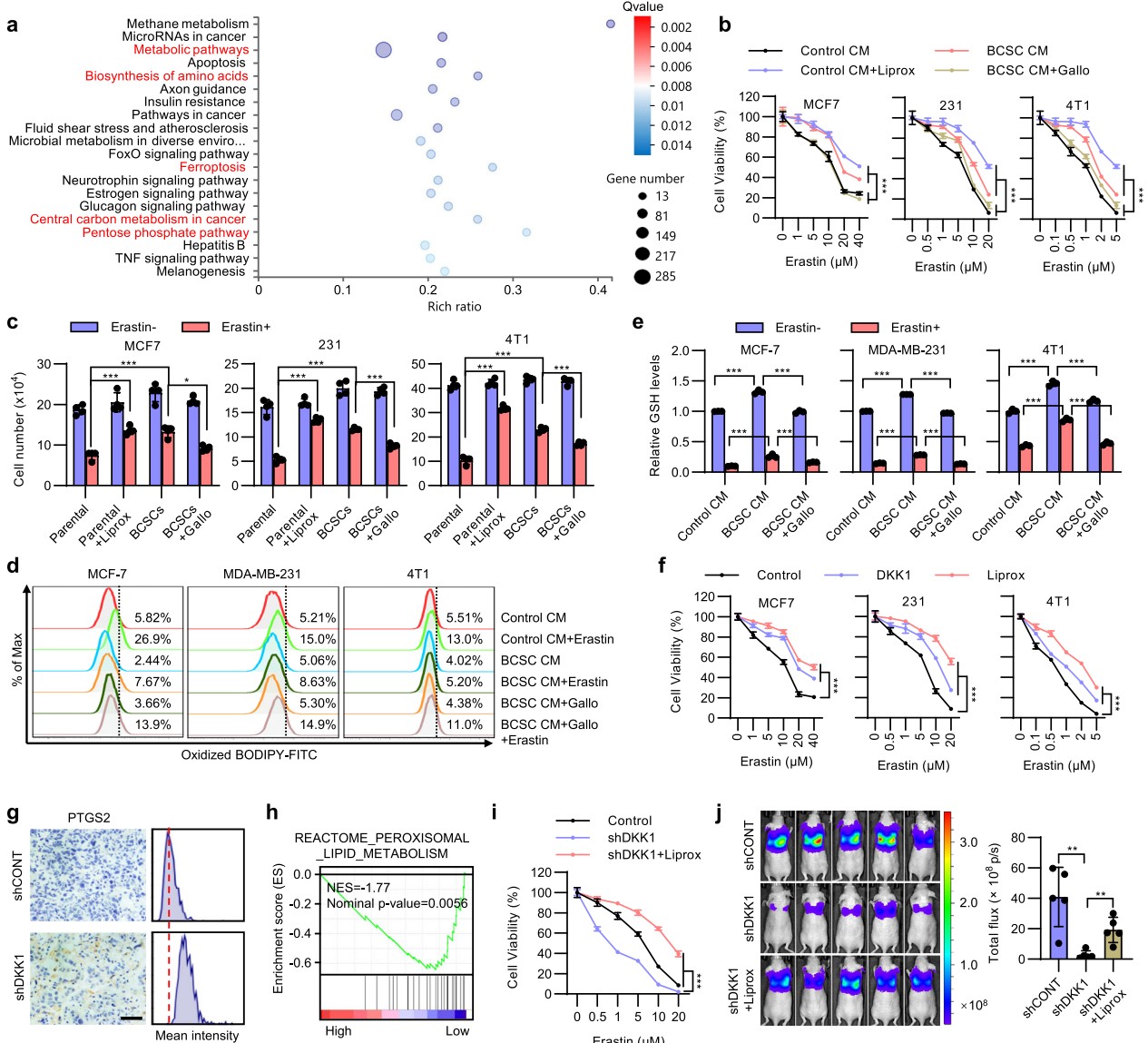

**Fig. 5 BCSC secretome confers ferroptosis resistance. a** KEGG analysis of the most enriched pathways in MCF-7 cells cultured with CM from BCSCs or parental cells. **b** Cell viability of MCF-7, MDA-MB-231 or 4T1 cells cultured with respective control CM ± 1 μM Liproxstatin-1 or BCSC CM ± 5 μM Gallocyanine and treated with a graded concentration of Erastin. **c** MCF-7, MDA-MB-231 or 4T1 cells were co-cultured with respective parental cells ± 1 μM Liproxstatin-1 or BCSCs ± 5 μM Gallocyanine and treated with 10 μM (MCF-7), 5 μM (MDA-MB-231), and 1 μM (4T1) Erastin, respectively. Cell viability was determined by cell count assay. **d** Lipid ROS in MCF-7, MDA-MB-231, or 4T1 cells cultured with respective control CM or BCSC CM for 48 h in the presence or absence of 5 μM Gallocyanine and treated with 10 μM (MCF-7), 5 μM (MDA-MB-231), and 1 μM (4T1) Erastin, respectively. **e** Relative GSH levels in MCF-7, MDA-MB-231, or 4T1 cells cultured with respective control CM or BCSC CM ± 5 μM Gallocyanine and treated with Erastin for 48 h. **f** Cell viability of MCF-7, MDA-MB-231, or 4T1 cells were cultured with 100 ng/ml recombinant DKK1 or 1 μM Liproxstatin-1 and treated with a graded concentration of Erastin for 48 h. **g** IHC staining of PTGS2 in lung metastases derived from MDA-MB-231-shCONT or -shDKK1 cells. Scale bar: 50 μm. **h** GSEA analysis of the enrichment of peroxisomal lipid metabolism in TCGA breast cancer patients with different DKK1 levels. **i** Cell viability of MDA-MB-231-shCONT or -shDKK1 cells ± 1 μM Liproxstatin-1 and treated with a graded concentration of Erastin for 48 h. **j** BLI of nude mice intravenously injected with $1 \times 10^6$ MDA-MB-231-shCONT or -shDKK1 cells and treated with vehicle or Liproxstatin-1 as indicated. Results are shown as mean ± S.D. *$P < 0.05$; **$P < 0.01$; ***$P < 0.001$; ns not significant (Two-way ANOVA test in (**b**, **f**, **i**) unpaired two-tailed Student's $t$ test in (**j**), others one-way ANOVA followed by Tukey's multiple comparison). Source data are provided as a Source Data file.

whereas DKK1 inhibitor addition to BCSC CM significantly ameliorated this resistance (Fig. 5b). Another ferroptosis inducer RSL3 was further applied and a similar effect was observed (Supplementary Fig. 6b). Similar results were also observed in cells co-cultured with the respective BCSCs (Fig. 5c). We further cultured these cells in the presence of CM from BCSCs stably transfected with shCONT or shDKK1. Consistently, CM from

DKK1-depleted BCSCs failed to protect cells against Erastin or RSL3 induced cell death, as compared to that from shCONT BCSC CM (Supplementary Fig. 6c, d). Ferroptosis is accompanied by increased lipid peroxidation, which is assayed using the fluorescent probe BODIPY-C11[34]. Flow cytometric staining with BODIPY-C11 showed that while mammary carcinoma cells cultured with BCSC CM exhibited lower lipid ROS levels compared

to cells cultured with control CM, concurrent treatment with Gallocyanine rescued this effect (Fig. 5d). Moreover, Erastin-induced accumulation of lipid ROS was further decreased in cells cultured with BCSC CM but not with BCSC CM with Gallocyanine addition (Fig. 5d). As cellular GSH plays a crucial role in ferroptosis[35], we also observed that the cellular GSH levels were significantly higher in cells cultured with BCSC CM (Fig. 5e). Furthermore, BCSC CM also rescued the Erastin-induced depletion of GSH, while Gallocyanine treatment significantly abrogated the effects observed above (Fig. 5e). These findings suggest a potential role of BCSC-secreted DKK1 in facilitating breast cancer cell evasion from ferroptosis.

For further verification, MCF-7, MDA-MB-231 and 4T1 cells were cultured with recombinant DKK1 and subsequently treated with graded concentrations of Erastin or RSL3. Similar to BCSC CM, DKK1 supplementation also protected cells from Erastin or RSL3 induced cell death (Fig. 5f and Supplementary Fig. 6e). Similarly, lipid ROS levels were decreased while GSH levels were increased upon DKK1 treatment (Supplementary Fig. 6f, g). DKK1 also rescued the Erastin-induced lipid ROS accumulation or GSH depletion in these cells (Supplementary Fig. 6f, g). Interestingly, Erastin treatment also increased DKK1 levels (Supplementary Fig. 6h). To assess whether DKK1-modulated cancer cell ferroptosis in vivo, IHC staining of PTGS2, a downstream marker of ferroptosis[35], was performed in lung metastases. Indeed, lung metastases derived from MDA-MB-231-shDKK1 cells exhibited higher PTGS2 levels compared to those from -shCONT cells (Fig. 5g). In addition, treatment with WAY262611 or Gallocyanine significantly increased the PTGS2 levels in the MDA-MB-231 derived lung metastases (Supplementary Fig. 6i). Furthermore, gene set enrichment analysis (GSEA) suggested that breast cancer patients with higher DKK1 levels showed lower enrichment of lipid peroxisomal metabolism associated genes (Fig. 5h). To determine if DKK1-modulated ferroptosis contributed to metastatic outgrowth, we first treated DKK1-depleted MDA-MB-231 cells with liproxstatin-1. Whereas DKK1 depletion sensitized cells to Erastin, liproxstatin-1 partially rescued this effect (Fig. 5i). Again, while DKK1 depletion in MDA-MB-231 cells largely abolished the metastatic burden in vivo, liproxstatin-1 potently rescued this effect (Fig. 5j). Consistently, liproxstatin-1 also reversed DKK1 depletion-dependent increased expression of the ferroptosis marker PTGS2 in the lung metastases (Supplementary Fig. 6j). The diminished percentage of proliferative Ki-67 positive cells in DKK1-depleted lung metastases was also reversed by liproxstatin-1 (Supplementary Fig. 6k). Thus, these results demonstrate that DKK1 promoted tumor metastasis by protecting cells against ferroptosis.

**DKK1 promotes SLC7A11 expression**. To define the mechanism by which BCSC-secreted DKK1 protects breast cancer from ferroptosis, the transcriptomes of MCF-7 cells cultured with CM from BCSCs or parental cells were analyzed by RNA-seq analysis and the significantly altered genes involved in ferroptosis were determined (Fig. 6a). SLC7A11, a component of the cysteine-glutamate transporter system $xc^-$ that plays critical roles in ferroptosis[36], was observed as one of the most upregulated genes in cells cultured with BCSC CM (Fig. 6a), which was further validated by qRT-PCR (Supplementary Fig. 7a). Consistent with previous reports[37], SLC7A11 depletion in MCF-7 and MDA-MB-231 cells resulted in reduced cell viability in the presence of Erastin or RSL3 (Supplementary Fig. 7b–d). The protein levels of SLC7A11 were also observed to be increased in MCF-7, MDA-MB-231 and 4T1 cells cultured with BCSC CM compared to control CM, while Gallocyanine supplementation in BCSC CM diminished this effect (Fig. 6b). We further demonstrated that

DKK1-regulated SLC7A11 expression by supplementing recombinant DKK1 in the medium (Fig. 6c). Consistently, DKK1 depletion was observed to reduce SLC7A11 expression in MDA-MB-231 derived metastases (Fig. 6d). To ascertain whether SLC7A11 is required for the BCSC secretome to protect cancer cells against ferroptosis, SLC7A11 was depleted by shRNA in MDA-MB-231 cells. BCSC CM protected parental control cells but not SLC7A11 depleted cells against Erastin-induced cell death (Fig. 6e). Breast cancer clinical data also showed that higher SLC7A11 expression was correlated with worse survival outcomes in lymph node-positive/metastatic patients but not in the lymph node-negative or overall patient population (Supplementary Fig. 7e).

The mechanism by which DKK1 upregulates SLC7A11 was also investigated. STAT3 was reported to bind to the SLC7A11 promoter region and repress its expression[38], whereas the WNT/β-CATENIN pathway increases STAT3 activity[39]. We therefore determined if DKK1 utilizes the β-CATENIN-STAT3 axis to modulate SLC7A11 expression. DKK1 supplementation led to decreased β-CATENIN levels and reduced STAT3 activation (Supplementary Fig. 7f). In contrast, DKK1 depletion resulted in increased STAT3 activity and decreased SLC7A11 levels, which was reversed by STAT3 inhibitor treatment (Fig. 6f). We further determined whether DKK1 protection of cells from ferroptosis is β-CATENIN dependent. Supplemental DKK1 was shown to be sufficient to promote Erastin resistance in parental cells but not in β-CATENIN depleted cells (Supplementary Fig. 7g).

For in vivo studies, we injected the luciferase-labeled MDA-MB-231 or 4T1 cells via the tail vein of host mice to form lung macro-metastases. Antibiotic-resistant cancer cell lines (designated as 231 LM or 4T1 LM, respectively) were subsequently retrieved from lung tissue. Consistent with elevated DKK1 levels in lymph node metastases in breast cancer (Fig. 3c), 231 LM and 4T1 LM cells exhibited elevated DKK1 levels compared to their respective parental cells (Fig. 6g). Interestingly, increased SLC7A11 levels were also persistently observed in 231 LM and 4T1 LM cells (Fig. 6g), suggesting resistance to ferroptosis is enhanced in successful lung metastases. Indeed, 231 LM and 4T1 LM cells showed decreased lipid ROS and increased GSH levels compared to the respective control cells (Fig. 6h, i), whereas 231 LM and 4T1 LM cells exhibited enhanced resistance to Erastin or RSL3 treatment compared to the respective controls (Fig. 6j and Supplementary Fig. 7h). Gallocyanine treatment re-sensitized 231 LM and 4T1 LM cells to Erastin or RSL3 (Fig. 6j and Supplementary Fig. 7h). To determine whether Gallocyanine sensitized cancer cells to ferroptosis during the metastatic process, we pretreated 4T1 cells with 1 μM Erastin or vehicle for 48 h, and then intravenously injected the cells into BALB/c mice. The mice were further treated with Gallocyanine or vehicle for 2 weeks. Single treatment with Erastin or Gallocyanine exhibited moderate effects in inhibiting metastasis (Fig. 6k). In contrast, combined treatment with Erastin and Gallocyanine exhibited greater efficacy in abrogating cancer metastasis (Fig. 6k). Similarly, combined treatment resulted in much higher levels of the ferroptosis marker PTGS2 and lower levels of proliferative marker Ki-67 in lung metastases compared to that treated with Erastin or Gallocyanine alone (Supplementary Fig. 7i, j).

**β-CATENIN activity is required for the secretion of DKK1**. The mechanism underlying the increased secretion of DKK1 in BCSCs was further investigated. It has been reported that DKK1 is a transcriptional target of β-CATENIN[40]. In line with previous reports, β-CATENIN protein levels were significantly higher in BCSCs than parental cells[41] (Fig. 7a). It was further observed that DKK1 levels were downregulated in β-CATENIN depleted MCF-

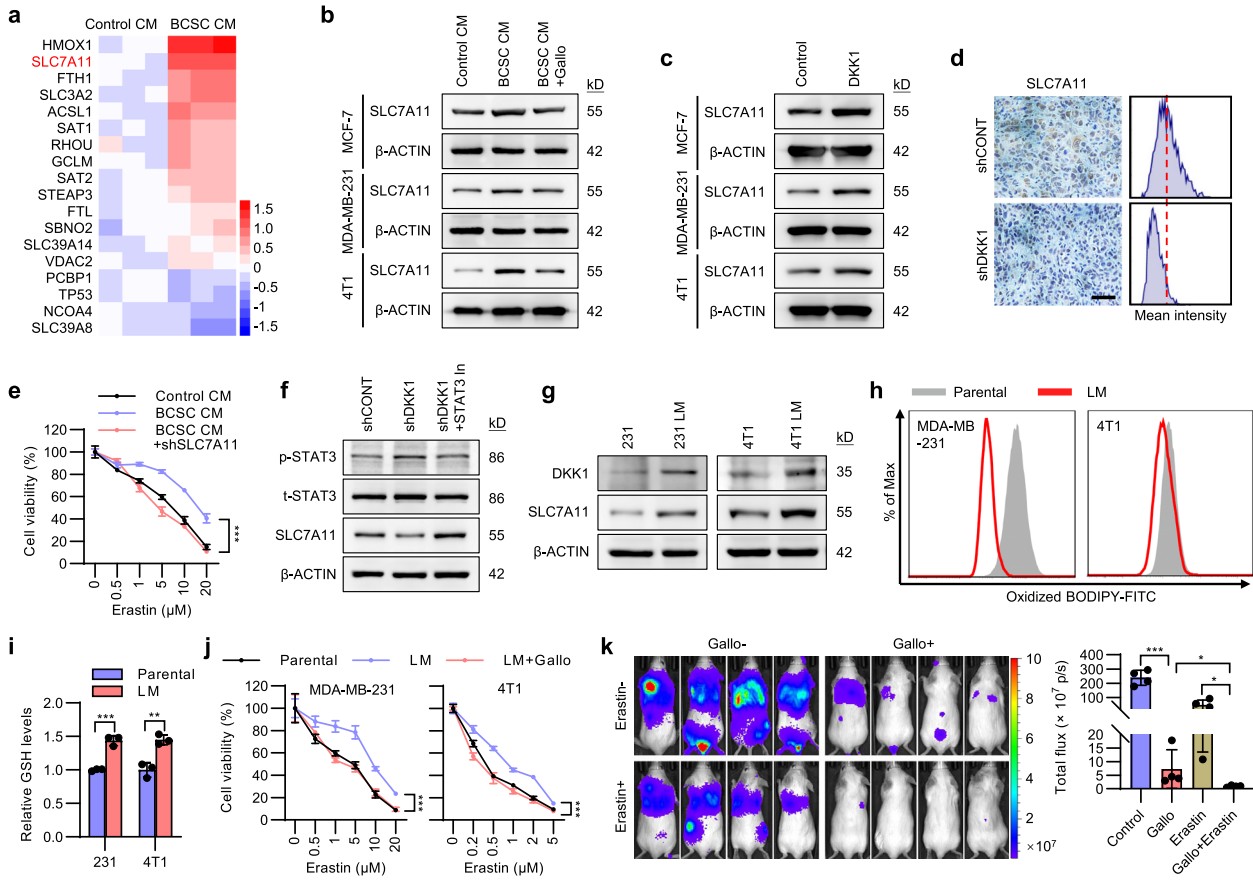

**Fig. 6 DKK1 promotes SLC7A11 expression. a** Heatmap represents the significantly altered ferroptosis related genes in MCF-7 cells cultured with CM from BCSCs or parental cells. **b** Immunoblot assessment of SLC7A11 protein levels. Cells were cultured with the CM obtained from respective parental cells or mammosphere-enriched BCSCs and treated with Gallocyanine (5 μM) or vehicle. **c** Immunoblot assessment of SLC7A11 protein levels in cells treated with vehicle or 100 ng/ml DKK1. **d** IHC staining of SLC7A11 in lung metastases derived from MDA-MB-231-shCONT or -shDKK1 cells. Scale bar: 50 μm. **e** Cell viability of MDA-MB-231-shCONT or -shSLC7A11 cells cultured with control CM or BCSC CM and treated with a graded concentration of Erastin for 48 h. **f** Immunoblot assessment of SLC7A11, phosphorylated and total STAT3 protein levels in MDA-MB-231-shCONT or -shDKK1 cells treated with or without 5 μM STAT3 Inhibitor. **g** Immunoblot assessment of DKK1 and SLC7A11 protein levels in the indicated cells. **h** Lipid ROS levels in the indicated cells. **i** The relative GSH levels in the indicated cells. **j** Cell viability of the parental or lung metastasis derived MDA-MB-231 or 4T1 cells treated with Gallocyanine and a graded concentration of Erastin for 48 h. **k** BLI of mice intravenously injected with $1 \times 10^5$ 4T1-luc cells. The mice were treated with Erastin, Gallocyanine or combined Erastin and Gallocyanine as indicated. Results are shown as mean ± S.D. *$P < 0.05$; **$P < 0.01$; ***$P < 0.001$; ns not significant (Two-way ANOVA test in (**e**, **j**) others unpaired two-tailed Student's *t* test). Source data are provided as a Source Data file.

7 or T47D cells (Fig. 7b). Consistently, the WNT/β-CATENIN signaling activator, CHIR99021, significantly increased β-CATENIN as well as DKK1 levels (Fig. 7c). In contrast, the WNT/β-CATENIN signaling inhibitor, XAV939, diminished the expression of β-CATENIN and DKK1. (Fig. 7c). BCSC CM derived from β-CATENIN depleted cells failed to decrease the CSC population (Supplementary Fig. 8a), supportive of the dependence on WNT/β-CATENIN signaling for DKK1 secretion. As expected, BCSC CM derived from β-CATENIN depleted cells failed to promote resistance to Erastin treatment (Fig. 7d).

It was next determined if the transcription of *DKK1* is regulated by the transcriptional activity of β-CATENIN signaling. Transcription factors of the TCF/LEF family are commonly involved in the transcriptional activation of downstream genes in WNT/β-CATENIN signaling[29], and the rVista 2.0 software predicted 2 conserved TCF4 binding sites in the promoter region of *DKK1* genes[42] (Fig. 7e). ChIP-sequencing data from the GEO database suggested the enrichments of TCF4 at the promoter region of *DKK1* (Fig. 7f). Additionally, WNT3a treatment increased the recruitment of β-CATENIN to the promoter region of *DKK1* (Fig. 7g). The promoter sequence of *DKK1* was cloned

into the pGL3-reporter plasmid, and the dependence on β-CATENIN for the activities of the *DKK1* reporters was demonstrated (Fig. 7h). Mutation of the second predicted TCF4 binding site of the *DKK1* promoter abolished the influence of β-CATENIN on reporter activity, but not the mutation of the first predicted site (Fig. 7h). The binding of TCF4 on the promoter of *DKK1* was further demonstrated by ChIP assay (Fig. 7i). ChIP analysis also showed increased recruitment of β-CATENIN to the promoter region of *DKK1* in BCSCs compared to parental cells (Fig. 7j). Therefore, WNT/β-CATENIN signaling in BCSCs controlled the secretion of DKK1 via transcriptional activation at its promoter.

## Discussion

Whereas the phenotypic plasticity of cancer cells in stem-like and differentiated states during propagation has been well recognized[4,5], the mechanism and functional roles of this phenomenon in cancer progression and therapy remain poorly understood. Much attention on the mechanism of phenotypic equilibrium has been paid to non-CSCs, which presumably form

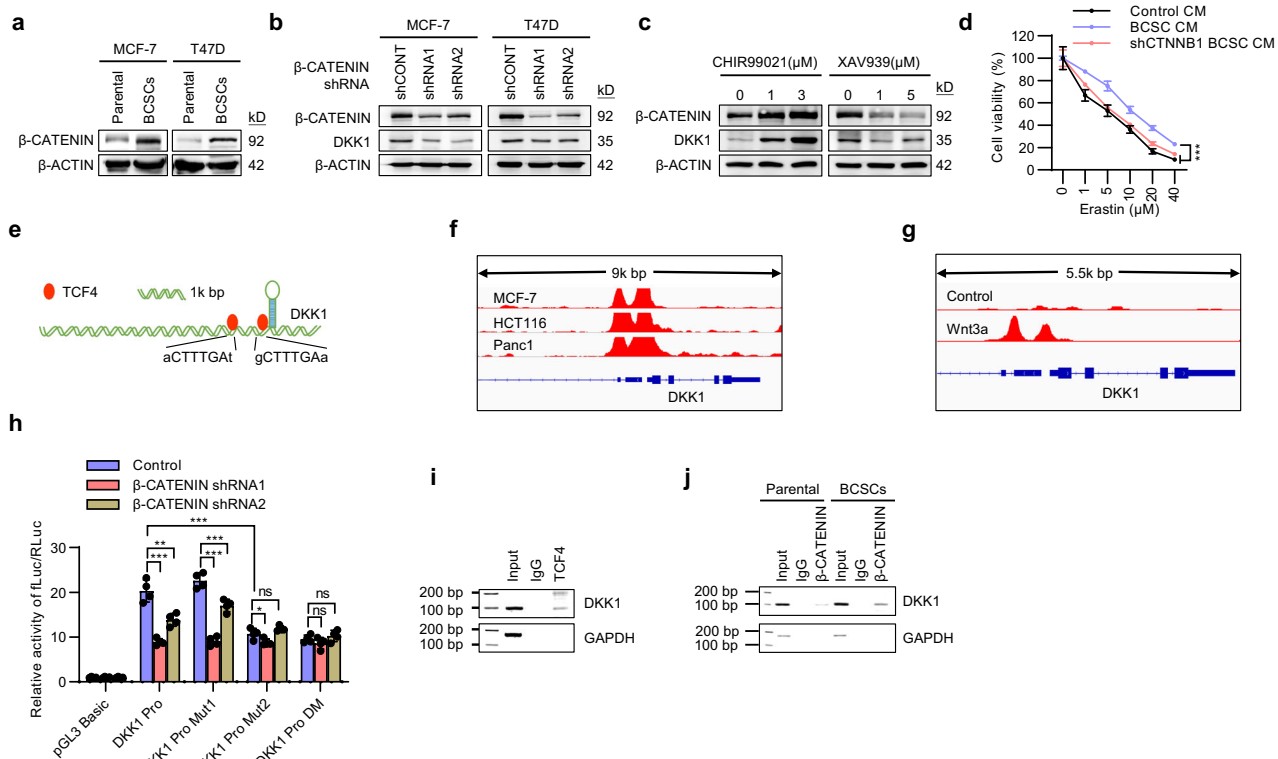

**Fig. 7 β-CATENIN regulates the transcription of DKK1. a** Immunoblot assessment of the protein levels of β-CATENIN in the matched mammosphere-enriched BCSCs and parental MCF-7 or T47D cells. **b** Immunoblot assessment of the protein levels of β-CATENIN and DKK1 in MCF-7 or T47D cells transfected with two different β-CATENIN shRNAs. **c** Immunoblot assessment of the protein levels of β-CATENIN and DKK1 in MCF-7 cells treated with CHIR99021 or XAV939. **d** Cell viability of MCF-7 cells cultured with the CM derived from parental cells or mammosphere-enriched shCONT BCSCs or shCTNNB1-BCSCs and treated with a graded concentration of Erastin for 48 h. **e** Schematic representation of the predicted TCF4 binding sites in the DNA promoter region of *DKK1* based on rVista 2.0 software. **f** ChIP-sequencing data shows the enrichments of TCF4 around the promoter region of *DKK1* in MCF-7, HCT116, and Panc1 cells (GSE31477). **g** ChIP-sequencing data shows the enrichments of β-CATENIN around the promoter region of *DKK1* in cells treated with or without Wnt3a (GSE64758). **h** Regulation of wild-type or mutant (TCF4 binding site deleted) *DKK1* promoter activities by β-CATENIN were determined by luciferase reporter assay. Renilla luciferase activity as input control. **i** Binding of TCF4 to the *DKK1* promoter region in MCF-7 cells was examined by ChIP assay. **j** The binding of β-CATENIN to the *DKK1* promoter region in BCSCs or parental MCF-7 cells was examined by ChIP assay. Results are shown as mean ± S.D. *$P < 0.05$; **$P < 0.01$; ***$P < 0.001$; ns not significant (One-way ANOVA followed by Tukey's multiple comparison test in (**h**), others two-way ANOVA test). Source data are provided as a Source Data file.

the supportive niche for CSC maintenance by secreting growth factors or extracellular matrix components[5,6,43,44]. Although loss of the supportive niche composed of non-CSCs may partially explain the rapid differentiation of enriched CSCs, this study demonstrated an unexpected autochthonous mechanism in which BCSCs secreted DKK1 to restrict the stem cell pool and reduce stemness yet promote distant metastasis. It was further determined that BCSCs secreted DKK1 to diminish the stemness of breast cancer cells by inhibition of canonical WNT signaling. Successful establishment of metastases requires CSCs to disseminate from primary sites and colonize at secondary sites[11]. The evidence that CSCs maintain a quiescent state[14,45] suggests that disseminated CSCs are likely to require and acquire a differentiated phenotype to promote metastatic outgrowth. It was observed that BCSC-regulated differentiation in the metastatic sites was required for DTCs to exit from dormancy and subsequently achieve metastatic colonization, and elevated DKK1 expression was also pivotal for these processes. Thus, the data herein supported a role of BCSC-regulated subpopulation plasticity in facilitating disseminated cancer cell metastatic outgrowth by promoting DTC differentiation (Supplementary Fig. 8b).

Although highly migratory and invasive in vitro, SUM159 and 4TO7 cells remain non-metastatic or inefficiently-metastatic

in vivo[46,47]. As both cell lines exhibited limited DKK1 expression, it was surprising that co-injection of BCSCs, which tilted the equilibrium to a differentiated state in the accompanying parental cells, endowed efficient metastatic capacity to the accompanying SUM159 or 4TO7 cells. As the inefficiently-metastatic 4TO7 cells appear to be able to disseminate and seed in lungs but fail to establish proliferative colonies[46], the data herein strongly suggest that BSCS-secreted DKK1 exerts its role specifically at the post-extravasation stage. Furthermore, it was demonstrated that BCSC-secreted DKK1 specifically promoted micrometastatic expansion without appreciable effects on local invasion or lung seeding. Mechanistic analyses revealed that BCSC-secreted DKK1 promotes SLC7A11 expression, decreased lipid peroxidation and increased glutathione, leading to diminished tumor ferroptosis and enhanced cancer cell proliferation in lung metastases. As the microenvironment of lung metastases exhibits higher oxidative and ferroptotic stress compared to primary mammary or subcutaneous tumors[24], ferroptosis antagonism exerts little or no effect on subcutaneous tumor growth, but efficiently increases the metastatic disease burden in highly metastatic tumors[25]. Previous studies demonstrated that ALDH+ BCSCs were coupled with Ki-67+ proliferative states in cultured cells and primary cancers[48]. The data herein demonstrated that Ki-67 was preferentially

expressed in ALDH− cells in lung metastases. This contradiction may be due to the high ferroptotic stress in lung metastasis promoting the dormancy of CSCs and the proliferation of non-CSCs. Furthermore, ferroptosis appears to modulate metastatic disease burden in cancers with high-metastatic propensity but not in cancers that inefficiently metastasize[25]. Ferroptosis may therefore specifically decrease metastatic outgrowth in which post-extravasation metastatic colonization is rate-limiting, but not in which dissemination is a rate-limiting step. Consistently, it was observed that DKK1 selectively increased the metastatic burden of breast cancer without impacting primary tumor growth. The data herein also suggested that metastatic cancer cells in the lung may adapt to high ferroptotic stress and develop resistance to ferroptosis by inducing elevated DKK1 expression. The Dickkopf protein DKK2 has also been reported to be expressed in colorectal cancer stem cells and to modulate cancer progression[49]. DKK1 has been reported to modulate tumor metastasis by dictating the NK cells, macrophages or neutrophils in the tumor microenvironment[14,50]. Herein, a novel mechanistic insight was observed in which DKK1-regulated differentiation protected metastatic cells from ferroptosis at the post-extravasation stage to achieve metastatic colonization.

DTCs may survive, yet lay latent for long periods without perceived relapse[51]. A strategy to abrogate or limit the expansion potential of the latent micro-metastases may represent an opportunity to overcome metastatic mortality. Herein, DKK1 inhibitors potently diminished the outgrowth of metastatic cells and markedly extended the survival outcomes of host mice. Nevertheless, targeting DKK1 did not eradicate tumor cells completely. Combined treatment by use of a DKK1 inhibitor and a ferroptosis inducer further enhanced efficacy as DKK1 antagonism significantly sensitized metastatic cells to ferroptosis. Thus, further systematic evaluation of the efficacy of DKK1 inhibition alone or combined with ferroptosis inducers may be warranted to limit or overcome metastatic pathologies. In summary, this study demonstrated the contribution of CSC-regulated phenotypic plasticity to metastatic colonization and identified novel therapeutic approaches to limit metastatic outgrowth.

## Methods

**Ethics approval**. This study complies with all relevant ethical regulations. The study protocol was approved by Biomedical Ethics Committee of University of Science and Technology of China.

**Human subjects**. The specimens used in this study were collected from the First Affiliated Hospital of USTC (Hefei, Anhui, China). The specimens of paraffin sections used in this study were obtained in previous clinical diagnosis and treatment, which will not cause physical and mental suffering to the patients. The privacy and personal information of the patients will be protected, and the specimens that subjects have explicitly refused to use will not be used. A waiver of informed consent was granted. The clinical research protocol was approved by the Biomedical Ethics Committee of USTC (2020-P-054). The study is compliant with all relevant ethical regulations. A summary of the clinical information of the patients is provided in Supplementary Table 2.

**Cell lines**. If not specified otherwise, all cell lines used in this study were obtained from ATCC. Cells were cryopreserved soon upon receipt and continuously cultured for <2 months. MCF-7, T47D, BT474, and MDA-MB-231 cells have been authenticated by STR genotyping. No cross-contamination of other human cells was observed. The cell lines utilized are 100% matched with those of ATCC. Possible mycoplasma contamination of all cell lines in the laboratory is routinely and regularly monitored. SUM159 cell and the PDX were a kind gift from Suling Liu lab (Fudan University, China). HEK293T was from Dr. Ping Gao (USTC). 4TO7 was from Dr. Lianfeng Zhang (PUMC). MCF-7, MDA-MB-231, HEK293T and 4TO7 cells were cultured in DMEM (Gibco) supplemented with 10% fetal bovine serum (Gibco) at 37 °C and 5% CO$_2$. T47D, BT474 and 4T1 cells were cultured in RPMI 1640 (Gibco) supplemented with 10% fetal bovine serum (Gibco) at 37 °C and 5% CO$_2$. SUM159 cells were cultured in Ham's F12 (Gibco) supplemented with 5% fetal bovine serum (Gibco), 5 µg/ml insulin (Sigma) and 1 µg/ml hydrocortisone (Sigma).

**Mice**. All animal experiments were approved by the Institutional Animal Care and Use Committee, University of Science and Technology of China (USTCA-CUC1801035). The study is compliant with all relevant ethical regulations regarding animal research. The 5-week-old female BALB/c nude, BALB/c and NOD-SCID mice were purchased from SLAC laboratory animal (Shanghai, China). The NCG mice were purchased from Gempharmatech (Nanjing, China). The MMTV-PyMT mice were from Dr. Zhenye Yang (USTC). All mice were housed under the SPF environment with a 12 h light–dark cycle at 22–24 °C with 50–60% humidity. In MCF-7 xenograft models, a slow-release pellet containing 0.36 mg of 17β-estradiol (Innovative Research of America) was subcutaneously implanted into the back of nude mice prior to the tumor implantation. WAY262611 was administered i.v. once per day at the concentration of 15 mg/kg, Gallocyanine was administered every 2 days i.v. at the concentration of 10 mg/kg. For liproxstatin-1 treatment, the liproxstatin-1 was administered with daily i.p. at the concentration of 20 mg/kg, control animals received the vehicle. In vivo bioluminescent imaging was performed to determine the tumor incidence, growth and metastatic burden of luciferase-labeled cells. Mice were injected i.p. with 150 µg/g of D-luciferin (12 mg/ml in PBS) and imaged 10 min after injection using a PerkinElmer IVIS Spectrum system. The bioluminescent imaging was quantified using Living Image 4.5 software.

**NK cells depletion**. In vivo depletion of NK cells in nude mice was achieved by i.v. injection of 40 µl of anti-asialo-GM1 (Wako Chemicals) every 4 days. The NK-cell depletion regimen was performed 2 days before tail vein injection of MDA-MB-231-luc cells.

**Reagents**. For DKK1 Inhibitors, Gallocyanine was from Santa Cruz, WAY262611 was from TargetMol. STAT3 inhibitor III (WP1066) was from Santa Cruz. EGF and bFGF recombinant proteins were from PeproTech. The DKK1 recombinant protein was purchased from Abcam. Insulin was purchased from Sigma-Aldrich. The Wnt/β-catenin pathway inhibitor XAV939 and activator CHIR99021 were from TargetMol. The ferroptosis inducer Erastin and RSL3 were from MedChem Express. The liproxstatin-1 was from TargetMol. BODIPY-C11 (D3861) was from ThermoFisher. ALDEFLUOR™ Kit (Cat#01700) was from Stem Cell. ChIP Assay Kit (Cat#P2078) was from Beyotime. Human DKK1 Quantikine ELISA Kit (Cat#DKK100B) were from R&D System.

**Plasmids construction and transfection**. The DKK1 expression plasmid was a kind gift from Guohong Hu (Chinese Academy of Sciences, Shanghai). The shRNA plasmids of human genes were obtained from The RNAi Consortium (MISSION® TRC shRNA library, Sigma-Aldrich). For luciferase reporter plasmids of DKK1 promoters, the DNA fragments upstream of DKK1 gene carrying TCF4 binding sites were cloned into the pGL3-Basic plasmid (Promega). The mutant constructs were generated using the QuickChange II XL site-directed mutagenesis kit (Stratagene). The coding sequence of the firefly luciferase, DKK1 (HOMO) and DKK1 (Mus) genes were amplified and sub-cloned into the pSin vector to generate expressing plasmid. The sequences of shRNAs, primers for cloning and qRT-PCR are listed in Supplementary Table 3. The transfection was carried out using Lipofectamine 3000 (Invitrogen).

**Lentivirus production and transduction**. The pSin-luciferase, pSin-DKK1 and shRNAs viruses were generated by transfection of the constructs together with pMD2.G and psPAX2 into HEK293T cells using calcium phosphate. 14 h after the transfection, the medium was replaced with preheated fresh medium. The virus particles were harvested 24 and 48 h later, filtered by 0.45 µm filter unit (Millipore). Cells were transduced with recombinant lentivirus with 10 µg/ml polybrene for 48 h and then selected by puromycin for 1 week.

**Mammosphere culture**. 5000 cells were seeded into each well of 6-well plates precoated with poly (2-hydroxyethyl metacrylate) (Polyhema; Sigma) which prevents the cells from attaching to the surface. Cells were cultured in Dulbecco's modified Eagle's medium (DMEM)/F12 (Gibco) supplemented with B27 (1:50; Gibco), bovine serum albumin (0.4%; Biofroxx), EGF (20 ng/ml; PeproTech), bFGF (20 ng/ml; PeproTech), insulin (5 µg/ml; Sigma), penicillin–streptomycin (Sangon Biotech), L-glutamine (Gibco) for about 8 days to allow the generation of mammospheres.

**Conditioned medium and co-culture system**. For the BCSC-derived conditioned medium, the BCSCs isolated from flow cytometry sorting or mammosphere culturing were seeded with the regular medium in monolayer culture for 48 h. The medium from parallelly cultured parental cells was used as the control medium. The CM was centrifuged at 2000 × g for 3 min and filtered with a 0.22 µm filter unit (Millipore) to deplete any cell debris. In the Boyden co-culture system (3 µm pore filters; Corning), BCSCs were seeded on the upper chamber, and the same number of parental cells were seeded on the lower compartment. In the control setting, the same number of parental cells were used in both chambers.

**qRT-PCR and western blot**. The total RNA was prepared using TRIzol (Invitrogen). RNA was then converted to cDNA using the RevertAid first strand cDNA synthesis kit (Thermo Scientific). The SYBR Premix Ex Taq kit (Takara) was used to determine the expression levels, GAPDH served as input control. Real-time PCR was run on Stratagene MX3000P. The protein was extracted using RIPA lysis buffer. Western blot was imaged on ImageQuant LAS4000mini. The primers used were listed in Supplementary Table 3.

**Lipid peroxidation and GSH analysis**. Tumor cells were seeded in 6-well plates and cultured with the respective treatments for 48 h. For BODIPY-C11 staining, cells were incubated with 2 μM (MDA-MB-231 and 4T1 cells) or 10 μM (MCF-7 cells) BODIPY-C11 for 30 min at 37 °C. Cells were then collected by trypsinization and washed twice with PBS, then analyzed immediately with a flow cytometer (BD Biosciences). The FACS data was analysed by Flowjo 7.6. The GSH levels were analyzed using the GSH detection kit (Beyotime, S0053) and carried out following the manufacturer's instructions. The GSH levels of each sample were normalized to the respective protein concentration.

**Luciferase reporter assay**. MCF-7 cells were seeded at about 60% confluence in 24-well plates. For β-catenin reporter assay, 0.2 μg Super 8 × TOP flash plasmid was transfected into cells using lipofectamine 3000 (Invitrogen). For *DKK1* promoter reporter, 0.2 μg pGL3-Basic luciferase reporters were transfected into MCF-7 cells stable transfected with β-catenin shRNA or vector. pRL-TK plasmid was provided as an internal transfection control. The transfected cells were lysed 48 h later, and the luciferase activities were determined by the Dual-Luciferase® Reporter Assay System (Promega).

**ChIP assay**. Chromatin immunoprecipitation was performed using the ChIP Assay kit (Beyotime) and carried out following the manufacturer's instructions. DNA enrichment was assessed by PCR using PrimeStar HS DNA Polymerase (Takara). The primers used are listed in Supplementary Table 3.

**Statistics and reproducibility**. Figure 1a–h, $n = 3$ biologically independent samples. The experiments were performed three times with similar results. b ns $p = 0.2902$; ***$p = 0.0005$, one-way ANOVA followed by Tukey's multiple comparison test, $F = 48.03$. c ns $p = 0.3079$; ***$p = 0.001$, one-way ANOVA followed by Tukey's multiple comparison test, $F = 37.26$. e *$p = 0.0385$; ***$p = 6.24e−6$, unpaired two-tailed Student's $t$ test. f **$p = 0.0016$; ***$p = 0.0003$, unpaired two-tailed Student's $t$ test. g *$p = 0.013$; ***$p = 0.00017$, unpaired two-tailed Student's $t$ test. h left **$p = 0.0072$, right **$p = 0.0013$, unpaired two-tailed Student's $t$ test. j, k $n = 8$ biologically independent samples. The experiments were performed one time. ***$p = 2.72e−7$.

Figure 2a, the experiments were performed three times with similar results. b $n = 3$ biologically independent samples. The experiments were performed three times with similar results. **$p$ (upper) = 0.0017, ***$p$ (medium) = 0.00041, ***$p$ (lower) = 5.42e−5, one-way ANOVA followed by Tukey's multiple comparison test, $F = 117.9$. c right, $n = 3$ biologically independent samples. **$p$ (DKK1) = 0.0012, *$p$ (DKK2) = 0.0353, *$p$ (WIF1) = 0.0286, ns for $p$ (DKK3) = 0.22, $p$ (DKK4) = 0.2916, $p$ (SOST) = 0.1686, $p$ (SFRP1) = 0.7868, $p$ (SFRP2) = 0.8530, unpaired two-tailed Student's $t$ test. d the experiments were performed three times with similar results. e $n = 4$ biologically independent samples. The experiments were performed three times with similar results. ***$p$ (MCF-7) = 3.89e−8, $p$ (T47D) = 1.85e−10, unpaired two-tailed Student's $t$ test. f $n = 4$ biologically independent samples. The experiments were performed three times with similar results. ***$p$ (MCF-7 left) = 0.00023, ***$p$ (MCF-7 median) = 0.00042, ***$p$ (MCF-7 right) = 0.0002, ***$p$ (T47D left) = 7.13e−5, ***$p$ (T47D median) = 0.00033, ***$p$ (T47D right) = 0.0004, one-way ANOVA followed by Tukey's multiple comparison test, $F$ (MCF-7) = 63.49, $F$ (T47D) 157.7. g $n = 3$ biologically independent samples. The experiments were performed three times with similar results. ***$p$ (MCF-7) = 0.0002, **$p$ (MCF-7 upper) = 0.0029, **$p$ (MCF-7 lower) = 0.002. ***$p$ (T47D left) = 0.00005, ***$p$ (T47D right) = 0.0004, ***$p$ (T47D lower) = 0.0003, one-way ANOVA followed by Tukey's multiple comparison test, $F$ (MCF-7) = 20.8, $F$ (T47D) = 29.34. h, i $n = 8$ biologically independent samples. The experiments were performed one time. ***$p = 1.82e−6$, *$p = 0.0313$.

Figure 3b, $n = 5$ biologically independent samples. The experiments were performed one time. **$p$ (left) = 0.0079, **$p$ (right) = 0.0079, unpaired two-tailed Student's $t$ test. c $n = 15$ biologically independent samples. The experiments were performed one time. ***$p = 0.0001$, paired two-tailed Student's $t$ test. d The experiments were performed three times with similar results. e $n = 5$ biologically independent samples. The experiments were performed one time. *$p = 0.0481$, unpaired two-tailed Student's $t$ test. f $n = 5$ biologically independent samples. The experiments were performed one time. g $n = 5$ biologically independent samples. The experiments were performed one time. *$p = 0.032$, unpaired two-tailed Student's $t$ test. h $n = 5$ biologically independent samples. The experiments were performed one time. **$p$ (left) = 0.0028, **$p$ (right) = 0.0014, unpaired two-tailed Student's $t$ test. i $n = 10$ biologically independent samples. The experiments were performed one time. ***$p$ (left) = 3.72e−11, ***$p$ (median) = 1.18e−8, ***$p$ (right) = 7.75e−12, unpaired two-tailed Student's $t$ test. j $n = 5$ biologically

independent samples. The experiments were performed one time, *$p = 0.0399$, unpaired two-tailed Student's $t$ test.

Figure 4a–c, $n = 5$ biologically independent samples. The experiments were performed one time. a **$p$ (upper) = 0.0079, **$p$ (lower) = 0.0079, unpaired two-tailed Student's $t$ test. c ***$p$ (upper) = 0.000992, ***$p$ (lower) = 6.57e−6, unpaired two-tailed Student's $t$ test. d $n = 11$ biologically independent samples. The experiments were performed one time. ***$p$ (upper) = 6.86e−12, ***$p$ (upper) = 2.72e−9, one-way ANOVA followed by Tukey's multiple comparison, $F = 139$. e $n = 10$ biologically independent samples. The experiments were performed one time. ***$p$ (CDKN1B upper) = 0.000298, ***$p$ (CHKN1B lower) = 5.86e−7, ***$p$ (CHEK1 upper) = 0.000144, ***$p$ (CHEK1 lower) = 1.35e −6, ***$p$ (TGFB2 upper) = 3.55e−13, ***$p$ (TGFB2 lower) = 3.32e−13, one-way ANOVA followed by Tukey's multiple comparison, $F$ (CDKN1B) = 233.1, F (CHEK1) = 100.1, $F$ (TGFB2) = 160.9. f $n = 20$ biologically independent samples. The experiments were performed one time. g, h $n = 27$ biologically independent samples. The experiments were performed one time. i–k $n = 5$ biologically independent samples. The experiments were performed one time. k **$p = 0.0248$, unpaired two-tailed Student's $t$ test.

Figure 5a, $n = 3$ biologically independent samples. The experiments were performed one time. b $n = 4$ biologically independent samples, the experiments were performed three times with similar results, ***$p$ (MCF-7 Control CM vs. Comtrol + Liprox) = 2.21e−12, ***$p$ (MCF-7 BCSC CM vs. BCSC CM + Gallo) = 1.83e−6, ***$p$ (231 Control CM vs. Comtrol + Liprox) < 1e−15, ***$p$ (231 BCSC CM vs. BCSC CM + Gallo) = 3.17e−13, ***$p$ (4T1 Control CM vs. Comtrol + Liprox) < 1e−15, ***$p$ (4T1 BCSC CM vs. BCSC CM + Gallo) = 1.94e−11, two-way ANOVA test. c $n = 4$ biologically independent samples, the experiments were performed three times with similar results, ***$p$ (MCF-7 upper) = 0.000234, ***$p$ (MCF-7 lower) = 9.77e−5, *$p$ (MCF-7) = 0.013, ***$p$ (231 left) = 1.44e−10, ***$p$ (231 median) = 2.78e−8, ***$p$ (231 right) = 0.000467, ***$p$ (4T1 left) = 1.8e−14, ***$p$ (4T1 median) = 5.44e−11, ***$p$ (4T1 right) = 7.76e−5, one-way ANOVA followed by Tukey's multiple comparison, $F$ (MCF-7) = 53.97, F (MDA-MB-231) = 53.94, $F$ (4T1) = 85.61. d $n = 3$ biologically independent samples, the experiments were performed three times with similar results. e, f $n = 3$ biologically independent samples, the experiments were performed three times with similar results. e ***$p$ (MCF-7 left upper) = 4.86e−9, ***$p$ (MCF-7 left lower) = 9.65e−6, ***$p$ (MCF-7 right upper) = 3.82e−9, ***$p$ (MCF-7 right lower) = 0.000878, ***$p$ (231 left upper) = 2.6e−14, ***$p$ (231 left lower) = 3.81e−13, ***$p$ (231 right upper) = 2.6e −14, ***$p$ (231 right lower) = 1.78e−13, ***$p$ (4T1 left upper) = 6.42e−9, ***$p$ (4T1 left lower) = 1.38e−8, ***$p$ (4T1 right upper) = 1.25e−6, ***$p$ (4T1 right lower) = 3.86e−8, one-way ANOVA followed by Tukey's multiple comparison, $F$ (MCF-7) = 1825, $F$ (MDA-MB-231) = 52650, $F$ (4T1) = 493. f ***$p$ (MCF-7 DKK1) = 7.5e−14, ***$p$ (MCF-7 Liprox) < 1e−15, ***$p$ (231 DKK1) = 1.27e−11, ***$p$ (231 Liprox) < 1e−15, ***$p$ (4T1 DKK1) < 1e−15, ***$p$ (4T1 Liprox)<1e −15, two-way ANOVA test. g $n = 5$ biologically independent samples. The experiments were performed one time. i $n = 4$ biologically independent samples, the experiments were performed three times with similar results. ***$p$ (Control vs. shDKK1) < 1e−15, ***$p$ (shDKK1 vs. shDKK1 + Liprox) < 1e−15, two-way ANOVA test. j $n = 5$ biologically independent samples, the experiments were performed one time. **$p$ (left) = 0.0025. **$p$(right) = 0.0029, unpaired two-tailed Student's $t$ test.

Figure 6a, $n = 3$ biologically independent samples, the experiments were performed one time. b, c the experiments were performed three times with similar results. d $n = 5$ biologically independent samples, the experiments were performed one time. e $n = 4$ biologically independent samples, the experiments were performed three times with similar results, ***$p$ (Control CM vs. BCSC CM) = 2.4e−12, ***$p$ (BCSC CM vs. BCSC CM + Liprox) < 1e−15, two-way ANOVA test. f, g the experiments were performed three times with similar results. h $n = 3$ biologically independent samples, the experiments were performed three times with similar results. i $n = 3$ biologically independent samples, the experiments were performed three times with similar results, ***$p = 0.000566$, **$p = 0.0038$, unpaired two-tailed Student's $t$ test. j $n = 4$ biologically independent samples, the experiments were performed three times with similar results, ***$p$ (231 Parental vs. LM) = 0.000152, ***$p$ (231 LM vs. LM + Gallo) = 2.31e−6, ***$p$ (4T1 Parental vs. LM) = 1.92e−11, ***$p$ (4T1 LM vs. LM + Gallo) = 2.9e−12, two-way ANOVA test. k $n = 4$ biologically independent samples, the experiments were performed one time, ***$p = 1.29e−6$, *$p$ (upper) = 0.0286, *$p$ (lower) = 0.034, unpaired two-tailed Student's $t$ test.

Figure 7a–c, the experiments were performed three times with similar results. d $n = 4$ biologically independent samples, the experiments were performed three times with similar results, ***$p$ (Control CM vs. BCSC CM) = 4.45e−5, ***$p$ (BCSC CM vs. shCTNNB1 BCSC CM) = 4.28e−6, two-way ANOVA test. h $n = 4$ biologically independent samples, the experiments were performed three times with similar results, ***$p$ (upper 1) = 8.5e−13, ***$p$ (upper 2) = 2.49e−6, ***$p$ (upper 3) = 2.41e−13, ***$p$ (upper 4) = 2.4e−13, **$p = 0.0012$, *$p = 0.0362$, one-way ANOVA followed by Tukey's multiple comparison test, $F = 119$. i–j the experiments were performed three times with similar results.

Supplementary Fig. 1a–i, $n = 3$ biologically independent samples. The experiments were performed three times with similar results. a, b See Fig. 1b. c ***$p = 1.75e−5$, unpaired two-tailed Student's $t$ test. d ***$p = 0.000635$, **$p$ (SOX2) = 0.0035, **$p$ (KLF4) = 0.0073, **$p$ (NANOG) = 0.0019, unpaired two-

tailed Student's $t$ test. e, f See Fig. 1e, g. g **$p$ (MCF-7) = 0.008, **$p$ (T47D) = 0.006, unpaired two-tailed Student's $t$ test. h ***$p$ (BT474) = 5.63e−5, ***$p$ (SUM159) = 0.00025, unpaired two-tailed Student's $t$ test. i ***$p$ (BT474) = 3.35e−5, ***$p$ (SUM159) = 2.71e−5, unpaired two-tailed Student's $t$ test. j $n = 4$ biologically independent samples. The experiments were performed three times with similar results. *$p$ = 0.0203, ns $p$ = 0.1343, unpaired two-tailed Student's $t$ test.

Supplementary Fig. 2a, b, The experiments were performed three times with similar results. c $n = 4$ biologically independent samples. The experiments were performed three times with similar results. ***$p$ = 0.0001, unpaired two-tailed Student's $t$ test. d The experiments were performed three times with similar results. e, f $n = 3$ biologically independent samples. The experiments were performed three times with similar results. e, ***$p$ (MCF-7) = 0.000144, ***$p$ (T47D) = 8.35e−5, unpaired two-tailed Student's $t$ test. f ***$p$ = 0.000784, *$p$ = 0.0102, unpaired two-tailed Student's $t$ test. g The experiments were performed three times with similar results. h See Fig. 2f. i $n = 3$ biologically independent samples. The experiments were performed three times with similar results. ***$p$ (MCF-7) = 0.000323, ***$p$ (T47D left) = 9.09e−6, ***$p$ (T47D right) = 0.00078, **$p$ (MCF-7) = 0.0028, **$p$ = 0.0011, one-way ANOVA followed by Tukey's multiple comparison test, $F$ (MCF-7) = 27.35, $F$ (T47D) = 197.6. j $n = 3$ biologically independent samples. The experiments were performed three times with similar results. ***$p$ (left) = 0.00038, ***$p$ (median) = 1.17e−5, ***$p$ (right) = 2.82e−5, **$p$ = 0.0014, one-way ANOVA followed by Tukey's multiple comparison test, $F$ (MCF-7) = 36.7, $F$ (T47D) = 129.5.

Supplementary Fig. 3a, The experiments were performed three times with similar results. b, c $n = 5$ biologically independent samples. The experiments were performed one time. ***$p$ = 0.0003, unpaired two-tailed Student's $t$ test. e–g $n = 3$ biologically independent samples. The experiments were performed three times with similar results. e ns $p$ (left) = 0.0813, ns $p$ (right) = 0.1232, unpaired two-tailed Student's $t$ test. g ***$p$ (left) = 5.69e−5, ***$p$ (right) = 0.000169, unpaired two-tailed Student's $t$ test. h The experiments were performed three times with similar results. i, j $n = 3$ biologically independent samples. The experiments were performed three times with similar results. i ***$p$ = 0.000147, unpaired two-tailed Student's $t$ test. j ns $p$ (left) = 0.4587, ns $p$ (right) = 0.2341, unpaired two-tailed Student's $t$ test. k $n = 5$ biologically independent samples. The experiments were performed one time. l $n = 10$ biologically independent samples. The experiments were performed one time. ns $p$ = 0.5352, two-way ANOVA test. m–p $n = 5$ biologically independent samples. The experiments were performed one time. m ns $p$ = 0.5678, unpaired two-tailed Student's $t$ test. n ns $p$ = 0.4959, unpaired two-tailed Student's $t$ test. o ns $p$ = 0.5042, unpaired two-tailed Student's $t$ test. q–s $n = 10$ biologically independent samples. The experiments were performed one time. q ***$p$ (KLF4) = 9.45e−6, ***$p$ (LIN28) = 0.000258, ***$p$ (NANOG) = 3.45e−9, ***$p$ (OCT3/4) = 4.3e−9, unpaired two-tailed Student's $t$ test. r ***$p$ (CCND1) = 0.00025, ***$p$ (CDK4) = 3.21e−7, unpaired two-tailed Student's $t$ test. s ns $p$ = 0.6493, unpaired two-tailed Student's $t$ test. t The experiments were performed three times with similar results.

Supplementary Fig. 4a, The experiments were performed three times with similar results. b–e $n = 5$ biologically independent samples. The experiments were performed one time. b *$p$ = 0.0399, unpaired two-tailed Student's $t$ test. d ns $p$ = 0.9506, unpaired two-tailed Student's $t$ test. e **$p$ = 0.0052, unpaired two-tailed Student's $t$ test. f The experiments were performed three times with similar results. g $n = 5$ biologically independent samples. The experiments were performed one time. *$p$ = 0.0159, unpaired two-tailed Student's $t$ test.

Supplementary Fig. 5a, b, The experiments were performed three times with similar results. c–e $n = 5$ biologically independent samples. The experiments were performed one time. c ns $p$ = 0.534, one-way ANOVA test, $F$ = 0.6614. f, g $n = 10$ biologically independent samples. The experiments were performed one time. f ***$p$ (KLF4 upper) = 3.31e−8, ***$p$ (KLF4 lower) = 2.93e−8, ***$p$ (LIN28 upper) = 6.3e−8, ***$p$ (LIN28 lower) = 5.61e−6, ***$p$ (NANOG upper) = 5.78e−9, ***$p$ (NANOG lower) = 3.67e−10, ***$p$ (OCT3/4) = 7.34e−7, ns $p$ = 0.9553, one-way ANOVA followed by Tukey's multiple comparison test, $F$ (KLF4) = 52.66, $F$ (LIN28) = 42.67, $F$ (NANOG) = 75.13, $F$ (OCT3/4) = 42.49. g ***$p$ (CCND1 upper) = 1.26e−6, ***$p$ (CCND1 lower) = 0.000285, ***$p$ (CDK4 upper) = 1.12e−8, ***$p$ (CDK4 lower) = 0.000155, one-way ANOVA followed by Tukey's multiple comparison test, $F$ (CCND1) = 30.35, $F$ (CDK4) = 49.16. h–j $n = 5$ biologically independent samples. The experiments were performed one time. h ***$p$ = 0.0003, unpaired two-tailed Student's $t$ test. j ***$p$ = 6.35e−6, unpaired two-tailed Student's $t$ test.

Supplementary Fig. 6a, $n = 3$ biologically independent samples. The experiments were performed three times with similar results. ***$p$ (MCF-7) = 0.000716, ***$p$ (4T1) = 2.66e−7, *$p$ = 0.018, unpaired two-tailed Student's $t$ test. b–e $n = 4$ biologically independent samples. The experiments were performed three times with similar results. b ***$p$ (MCF-7 Control CM vs. BCSC CM) = 4.8e −14, ***$p$ (MCF-7 BCSC CM vs. BCSC CM + Gallo) < 1e−15, ***$p$ (231 Control CM vs. BCSC CM) < 1e−15, ***$p$ (231 BCSC CM vs. BCSC CM + Gallo) = 6.9e −14, ***$p$ (4T1 Control CM vs. BCSC CM) = 6.54e−10, ***$p$ (4T1 BCSC CM vs. BCSC CM + Gallo) = 1.91e−8, two-way ANOVA test. c ***$p$ (MCF-7 Control CM vs. BCSC CM) = 9.18e−7, ***$p$ (MCF-7 BCSC CM vs. shDKK1 BCSC CM) = 1.29e−6, ***$p$ (231 Control CM vs. BCSC CM) = 1.33e−6, ***$p$ (231 BCSC CM vs. shDKK1 BCSC CM) = 4.65e−7, ***$p$ (4T1 Control CM vs. BCSC CM) = 2.71e−9, ***$p$ (4T1 BCSC CM vs. shDKK1 BCSC CM) = 3.32e−10, d, ***$p$ (MCF-7 Control CM vs. BCSC CM) = 1.37e−7, ***$p$ (MCF-7 BCSC CM vs.

shDKK1 BCSC CM) = 2.02e−7, ***$p$ (231 Control CM vs. BCSC CM) = 9.78e −10, ***$p$ (231 BCSC CM vs. shDKK1 BCSC CM) = 1.1e−5, ***$p$ (4T1 Control CM vs. BCSC CM) = 3.48e−11, ***$p$ (4T1 BCSC CM vs. shDKK1 BCSC CM) = 1.27e−12, e ***$p$ (MCF-7) = 1.63e−13, ***$p$ (231) = 9.71e−9, ***$p$ (4T1) = 6.83e−11, f, g $n = 3$ biologically independent samples. The experiments were performed three times with similar results. g ***$p$ (MCF-7 upper) = 5.1e−14, ***$p$ (MCF-7 lower) = 2.05e−8, ***$p$ (231 upper) = 1.76e−11, ***$p$ (231 lower) = 2.75e−7, ***$p$ (4T1 upper) = 4.93e−8, ***$p$ (4T1 lower) = 4.81e−7, one-way ANOVA followed by Tukey's multiple comparison test, $F$ (MCF-7) = 32634, $F$ (MDA-MV-231) = 22676, $F$ (4T1) = 953.1. h The experiments were performed three times with similar results. i–k $n = 5$ biologically independent samples. The experiments were performed one time. k ***$p$ (left) = 5.84e−7, ***$p$ (right) = 0.000162, one-way ANOVA followed by Tukey's multiple comparison test, $F$ = 55.63.

Supplementary Fig. 7a, $n = 3$ biologically independent samples. The experiments were performed three times with similar results. **$p$ = 0.0018, unpaired two-tailed Student's $t$ test. b The experiments were performed three times with similar results. c, d $n = 4$ biologically independent samples. The experiments were performed three times with similar results. c ***$p$ (MCF-7) = 2e−15, ***$p$ (231) < 1e−15, two-way ANOVA test. d ***$p$ (MCF-7) = 7.24e−10, ***$p$ (231)<1e−15, two-way ANOVA test. f The experiments were performed three times with similar results. g, h, $n = 4$ biologically independent samples. The experiments were performed three times with similar results. g ***$p$ (shCONT vs. shCONT + DKK1) = 5.53e−13, ***$p$ (shCONT vs. shCTNNB1) = 4.32e−7, two-way ANOVA test. h ***$p$ (231 Parental vs. LM) = 6.52e−12, ***$p$ (231 LM vs. LM + Gallo) = 2.36e−12, ***$p$ (4T1 Parental vs. LM) = 3.65e−9, ***$p$ (4T1 LM vs. LM + Gallo) < 1e−15, two-way ANOVA test. i $n = 4$ biologically independent samples. The experiments were performed one time. j $n = 4$ biologically independent samples. The experiments were performed one time. ***$p$ (left) = 3.68e−6; ***$p$ (right) = 0.0002; *$p$ = 0.0258, one-way ANOVA followed by Tukey's multiple comparison test, $F$ = 56.02.

Supplementary Fig. 8a, $n = 3$ biologically independent samples. The experiments were performed three times with similar results. *$p$ (left) = 0.0126, *$p$ (right) = 0.0117, one-way ANOVA followed by Tukey's multiple comparison test, $F$ = 12.3.

**Statistical analysis and reproducibility**. The Kaplan–Meier plotter database was used to analyze the effect of *DKK1* or *SLC7A11* on the survival of cancer patients. The database sources in Kaplan–Meier plotter include GEO, EGA, and TCGA. The IHC staining was analysed by HistoQuest tissue analysis software. Data were presented as mean ± SD (standard deviation), and GraphPad Prism (San Diego, CA) was used for the statistical analysis. The methods to determine statistical significance in each result were mentioned in the figure legend. All experiments were repeated at least three times. $p < 0.05$ was considered as statistically significant.

**Reporting summary**. Further information on research design is available in the Nature Research Reporting Summary linked to this article.

## Data availability
The RNA sequencing data has been deposited in GEO with the accession number: GSE156454. The ChIP-sequencing data used in this study was downloaded from GEO dataset: GSE31477 and GSE64758. The RNA sequencing data used in this study was downloaded from GEO dataset: GSE59653 and GSE80213 or TCGA dataset (http://www.cbioportal.org/). All other data supporting the findings of this study are available within the article and its Supplementary Files. Source data are provided with this paper.

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

## Acknowledgements

We thank Drs. Li Bai and Huafeng Zhang (USTC, China) for the technical support in flow cytometry. This work was supported by The National Natural Science Foundation of China (82173254, 81972470) to T.Z., Shenzhen Key Laboratory of Innovative Onco-therapeutics (ZDSYS20200820165400003) (Shenzhen Science and Technology Innovation Commission), China; Shenzhen Development and Reform Commission Subject Construction Project ([2017]1434), China; Universities Stable Funding Key Projects (WDZC20200821150704001); The Shenzhen Bay Laboratory, Oncotherapeutics (21310031), China; TBSI Faculty Start-up Funds, China to P.E.L., The National Natural Science Foundation of China (82103531) to M.W., and The National Natural Science Foundation of China (81702594) to H.Y.

## Author contributions

M.W., X.Z., P.E.L., and T.Z. conceived the project and designed the experiments. M.W., X.Z., W.Z., Y.S.C., X.L., and W.Q. collected most of the data. S.Li, X.H. and Y.P. provided the clinical samples. H.Y. supported the histochemical analysis. T.L. performed the hematological and blood biochemical detection. S.Liu. provided the PDX line and technical support. M.W., X.Z., W.Z., P.Q., M.Z., P.E.L. and T.Z. analyzed the results. M.W., X.Z., P.E.L., and T.Z. wrote the paper.

## Competing interests

The authors declare no competing interests.
