## [Peer Review File · Nature Communications]

Reviewers' Comments:

Reviewer #1:

Remarks to the Author:

In this manuscript the authors demonstrate a paracrine interaction between breast cancer stem cells (CSC's) and their differentiated progeny in a process mediated by DKK1 . They show in mouse models that CSC secreted DKK1 induces cell differentiation and tumor growth at metastatic sites. In addition DKK1 protected metastasizing cancer cells from ferroptosis via upregulation of SLC7A11. Treatment with a combination of ferroptosis inducer and DKK1 inhibitor reduced metastasis. In general, the experiments are well done and provide important new information on the paracrine regulation of breast CSC's. However, several important issues such as EMT/MET CSC plasticity need to be more thoroughly addressed as follows:

- 1) The authors claim that DKK1 induces growth at metastatic sites via induction of differentiation of CSCs. However, they fail to discuss the role of EMT/MET CSC transitions in these processes. One of the Coauthors previously demonstrated that breast CSC transition between a quiescent invasive EMT state characterized as CD44+/CD24- and a proliferative more epithelial CSC characterized by ALDH expression (Liu Stem Cell Reports 2:78 2014). The authors predominantly use ALDH expression as a marker of stemness. They do show in Ext fig 1a-1b that CD44+/CD24- cells have higher mammosphere generating capacity. However, they should also use these EMT CSC markers to examine EMT/MET CSC plasticity.
- 2) In fig 1c if 98% of cells are ALDH- why are there differences between the ALDH- cells and unsorted cells?
- 3) The authors should consider showing the Aldefluor data in bar graphs rather than flow cytometry figs. This would facilitate comparison of results.
- 4) Did the authors determine whether the effects of co injection of BCSC with SUM159 luc cells could be duplicated by pre treating the Sum159 Luc cells with conditioned medium from the CSC,s.
- 5) Fig 3g and Ext Fig 30 suggest that Ki67 was only expressed in ALDH neg cells. How do the authors reconcile this with their previous data that ALDH is expressed in proliferating epithelial CSC,s.
- 6) In fig 3k what subtypes of breast cancer are shown?
- 7) The Massague lab has demonstrated that DKK1 regulates metastatic latency via the suppression of NK cell activating ligands such as CD155 and ULBP2/4/5 resulting in a dormant state that is enforced by Natural Killer cells. The authors perform all studies in NOD/scid, nude, balb/c and MMTV-PyMT mice, all of which have natural killer cells present (though they are less functional in NODscid mice). As a result, the authors cannot rule out that Natural Killer cells are playing a significant role in the anti-metastasis effects of DKK1 inhibition. Additionally, a 2018 paper demonstrates that the ferroptosis inhibitor erastin promotes PBMC proliferation and differentiation into natural killer cells (doi: 10.1016/j.bbrc.2018.07.100). Experiments in NSG (nod scid gamma) mice that do not have NK cells, as well as depletion of NK cells in nude mice with Anti-asialo GM1 are recommended to answer this major question of whether NK cells are playing a role in the pro-metastasis properties of DKK1.
- 8) The authors should cite several related papers including Mai TT et al Nature Chemistry 2017 PMC589097 that demonstrates ferroptosis of EMT breast CSC and studies demonstrating a role of DKK proteins in paracrine regulation of CSC ie Shin iScience 2021 Apr15,24 (5).

Reviewer #2:

Remarks to the Author:

In the manuscript by Wu et al. "Cancer Stem Cell Regulated Phenotypic Plasticity Protects Metastasized Cancer Cells from Ferroptosis" the authors found that while co-implantation with breast cancer stem cells (BCSC) decreases the tumor-initiating capacity, it increases metastasis of accompanying cancer cells. Hereby, DKK1 was identified as the most abundant upregulated gene secreted by BCSCs. DKK1 inhibitors such as WAY262611 and Gallocyanine substantially prevented metastasis in various tumor and metastasis models. Furthermore, they suggested that DKK1 increases the expression of SLC7A11, therefore protecting metastasizing cancer cells from lipid peroxidation and subsequent ferroptosis. In addition, they present data showing that a combination therapy with a ferroptosis inducer (Erastin) and DKK1 inhibitor (Gallocyanine)

displayed a synergistic effect in preventing metastasis.

The authors support their hypothesis by using a comprehensive compilation of results, ranging from in vitro data to different genetic, syngeneic and human xenograft tumor models, including models of experimental lung metastasis and a clinically relevant PDX line. However, there are some major concerns regarding the detection of SLC7A11 and the use of Erastin, which should be addressed:

Figure 5. Liproxstatin-1 should be included to validate whether Erastin induces ferroptosis in the cell lines used. Based on Doll et al. Nature 2019 (Extended Data Fig. 7), the drug target of Erastin (SLC7A11 or xCT) is not expressed (or at very low levels) on MCF-7 or MDA-MB-231 cells. It is advisable to include Liproxstatin-1 in all or most experiments to prove whether cell death or other relevant observations are indeed caused by ferroptosis.

Figure 5d. It should be stated what was the concentration of Erastin used, and what was the incubation time

Figure 5i. It seems like Liproxstatin-1 only rescues the effect of DKK1 depletion in Erastin treated cells, but not the cell death caused by Erastin. How can this be explained?

l. 436: when 4T1 cells are pretreated for 48h with Erastin, according to Fig 5b, all cells should be dead, or did the authors use a sublethal concentration?

Figure 6. Many currently available antibodies against SLC7A11 are non-specific and unsuitable for determining SLC7A11 levels (Jiang et al Nat Rev Mol Cell Biol 2021). The authors use an antibody that detects a band size of 50 kD. Normally SLC7A11 should run at 35-40 kD. Using a validated antibody, it was shown that SLC7A11 is hardly expressed on MDA-MB-231 or MCF-7 cells (for reference check Doll et al. Nature 2019). Therefore, all data using the current antibody should be repeated using a suitable antibody.

Gallocyanine is a phenoxazine derivative; as phenoxazine is a privileged scaffold for ferroptosis inhibitors or radical-trapping antioxidants (RTA) (see Farmer et al JOC 2021), did the authors check whether Gallocyanine has potential RTA activity, for example at high concentrations?

REVIEWER COMMENTS

Reviewer #1 (Remarks to the Author):

In this manuscript the authors demonstrate a paracrine interaction between breast cancer stem cells (CSC's) and their differentiated progeny in a process mediated by DKK1. They show in mouse models that CSC secreted DKK1 induces cell differentiation and tumor growth at metastatic sites. In addition, DKK1 protected metastasizing cancer cells from ferroptosis via upregulation of SLCA11. Treatment with a combination of ferroptosis inducer and DKK1 inhibitor reduced metastasis. In general, the experiments are well done and provide important new information on the paracrine regulation of breast CSC's. However, several important issues such as EMT/MET CSC plasticity need to be more thoroughly addressed as follows:

Answer: We appreciate the positive and constructive comments of the reviewer. The revised manuscript is supplemented with further pertinent experimental data and controls. Please refer to the details below and also the changes made in the revised manuscript.

1) The authors claim that DKK1 induces growth at metastatic sites via induction of differentiation of CSCs. However, they fail to discuss the role of EMT/MET CSC transitions in these processes. One of the Coauthors previously demonstrated that breast CSC transition between a quiescent invasive EMT state characterized as CD44⁺/CD24⁻ and a proliferative more epithelial CSC characterized by ALDH expression (Liu Stem Cell Reports 2:78 2014). The authors predominantly use ALDH expression as a marker of stemness. They do show in Ext fig 1a-1b that CD44⁺/CD24⁻ cells have higher mammosphere generating capacity. However, they should also use these EMT CSC markers to examine EMT/MET CSC plasticity.

Answer: We thank the reviewer for the constructive suggestion. We have cultured MCF-7 and T47D cells with CM derived from BCSCs or parental cells and determined the BCSC population by using CD44/CD24. Consistently, the CD44⁺/CD24⁻ BCSC population was also decreased in MCF-7 or T47D cells cultured with BCSC CM compared to that of control CM. This data is now included in the manuscript as Extended Data Fig. 1g.

Extended Data Fig. 1g, FACS analysis of the proportion of CD44⁺CD24^{-/low} BCSCs in MCF-7 or T47D cells cultured with control CM or BCSC CM.

2) In fig 1c if 98% of cells are ALDH⁻ why are there differences between the ALDH⁻ cells and unsorted cells?

Answer: In Fig. 1b-1c, a slight increase of ALDH⁺ BCSCs was observed in T47D cells co-cultured with ALDH⁻ cells or cultured with CM derived from ALDH⁻ cells compared to that of the unsorted parental cells. This may be due to the gating strategy of the FACS sorting. As the ALDH⁺ cells are rare, all the ALDH⁺ cells are collected based on the DEAB control gate. However, only the weaker 50% of the ALDH⁻ cells were collected rather than the whole ALDH⁻ cell population (the gating strategy is shown below). Thus, the sorted ALDH⁻ cells may have a more differentiated phenotype compared to the unsorted cells.

The gating strategy of the FACS sorting

3) The authors should consider showing the Aldefluor data in bar graphs rather than flow cytometry figs. This would facilitate comparison of results.

Answer: As per the reviewer's suggestion, the Aldefluor data has been replaced by bar graphs in the main figures, and the FACS images are now shown in the extended data figures.

4) Did the authors determine whether the effects of co injection of BCSC with SUM159 luc cells could be duplicated by by pre treating the Sum159 Luc cells with conditioned medium from the CSCs.

Answer: We thank the reviewer for the suggestion for this experiment. SUM159-luc cells were cultured with BCSC CM for 48 hours and subsequently injected into the tail vein of nude mice. Lung metastasis was determined by BLI 30 days later. As shown below, although the result was not statistically significant, one out of five mice developed potent lung metastasis derived from the non-metastatic SUM159 cells cultured with BCSC CM. This may be due to a shorter period of culture in conditioned medium *in vitro* which might be insufficient to support the metastasis of the weak-metastatic SUM159 cells over a long-term. This data is now included in the manuscript as Extended Data Fig. 3c.

Extended Data Fig. 3c, BLI was performed on the metastatic burden of SUM159-luc cells cultured with control CM or BCSC CM for 48 hours.

5) Fig 3g and Ext Fig 30 suggest that Ki67 was only expressed in ALDH neg cells. How do the authors reconcile this with their previous data that ALDH is expressed in proliferating epithelial CSCs.

Answer: Liu and colleagues had previously demonstrated that the proliferation marker Ki-67 was preferentially expressed in ALDH+ cells in cultured cells and primary tumors¹. The data herein demonstrated that Ki-67 was only expressed in ALDH- cells in lung metastases. In this study, it was observed that the microenvironment of lung metastases exhibits much higher ferroptotic stress compared to primary mammary tumors², which promotes the dormancy of CSCs but the proliferation of non-CSCs, as the CSC population has been identified to be highly susceptible to ferroptosis-induced cell death³. Thus, Ki-67 was predominantly expressed in ALDH- cells in the lung metastases. We have now discussed this contradiction in the discussion section.

6) In fig 3k what subtypes of breast cancer are shown?

Answer: In figure 3k, all molecular subtypes of breast cancer were included, including Luminal A/B, HER2 amplified and TNBC. We have now included this information in the figure legend.

7) The Massague lab has demonstrated that DKK1 regulates metastatic latency via the suppression of NK cell activating ligands such as CD155 and ULBP2/4/5 resulting in a dormant state that is enforced by Natural Killer cells. The authors perform all studies in NOD/scid, nude, balb/c and MMTV-PyMT mice, all of which have natural killer cells present (though they are less functional in NODscid mice). As a result, the authors cannot rule out that Natural Killer cells are playing a significant role in the anti-metastasis effects of DKK1 inhibition. Additionally, a 2018 paper demonstrates that the ferroptosis inhibitor erastin promotes PBMC proliferation and differentiation into natural killer cells (doi: 10.1016/j.bbrc.2018.07.100). Experiments in NSG (nod scid gamma) mice that do not have NK cells, as well as depletion of NK cells in nude mice with Anti-asialo GM1 are recommended to answer this major question of whether NK cells are playing a role in the pro-metastasis properties of DKK1.

Answer: We thank the reviewer for the constructive suggestion. To determine whether NK cells are involved in the anti-metastatic effects of DKK1 inhibition, we

depleted BALB/c nude mice of NK cells by administration of polyclonal anti-asialo-GM1 antibody. Gallocyanine consistently abrogated lung metastases derived from tail vein injected MDA-MB-231 cells in these nude mice (Fig. 4k). We also injected 4T1 cells into the tail vein of NOD-Prkd^{cem26Cd52}Il2rg^{em26Cd22}/NjuCr1 (NCG) mice (no NK cells) and applied Gallocyanine or saline treatment. Gallocyanine also significantly abrogated the lung metastasis in these NCG mice (Extended Data Fig. 5j), suggesting an NK cells independent manner of DKK1-mediated metastasis. This new data is now incorporated into the revised manuscript.

Fig. 4k and Extended Data Fig. 5j, **k**, BLI of the metastatic burden of NK-cell-depleted BALB/c nude mice intravenously injected with 5×10^5 MDA-MB-231-luc cells. **j**, BLI of the metastatic burden of NCG mice intravenously injected with 1×10^5 4T1-luc cells. The mice were treated with vehicle or Gallocyanine.

8) The authors should cite several related papers including Mai TT et al Nature Chemistry 2017 PMC589097 that demonstrates ferroptosis of EMT breast CSC and studies demonstrating a role of DKK proteins in paracrine regulation of CSC ie Shin iScience 2021 Apr15,24 (5).

Answer: We thank the reviewer for the suggestion. The main conclusion of Mai TT et al Nature Chemistry on CSC and ferroptosis has now been discussed in the introduction section (ref. 23), and the role of DKK2 in colorectal CSCs (Shin et al. iScience 2021) has now been cited in the discussion section (ref. 49).

Reviewer #2 (Remarks to the Author):

In the manuscript by Wu et al. “Cancer Stem Cell Regulated Phenotypic Plasticity Protects Metastasized Cancer Cells from Ferroptosis” the authors found that while co-implantation with breast cancer stem cells (BCSC) decreases the tumor-initiating capacity, it increases metastasis of accompanying cancer cells. Hereby, DKK1 was identified as the most abundant upregulated gene secreted by BCSCs. DKK1 inhibitors such as WAY262611 and Galloyanine substantially prevented metastasis in various tumor and metastasis models. Furthermore, they suggested that DKK1 increases the expression of SLC7A11, therefore protecting metastasizing cancer cells from lipid peroxidation and subsequent ferroptosis. In addition, they present data showing that a combination therapy with a ferroptosis inducer (Erastin) and DKK1 inhibitor (Galloyanine) displayed a synergistic effect in preventing metastasis. The authors support their hypothesis by using a comprehensive compilation of results, ranging from in vitro data to different genetic, syngeneic and human xenograft tumor models, including models of experimental lung metastasis and a clinically relevant PDX line. However, there are some major concerns regarding the detection of SLC7A11 and the use of Erastin, which should be addressed:

Answer: We appreciate the positive and constructive comments of the reviewer. The revised manuscript is supplemented with further pertinent experimental data and controls. Please refer to the details below and also the changes made in the revised manuscript.

Figure 5. Liproxstatin-1 should be included to validate whether Erastin induces ferroptosis in the cell lines used. Based on Doll et al. Nature 2019 (Extended Data Fig. 7), the drug target of Erastin (SLC7A11 or xCT) is not expressed (or at very low levels) on MCF-7 or MDA-MB-231 cells. It is advisable to include Liproxstatin-1 in all or most experiments to prove whether cell death or other relevant observations are indeed caused by ferroptosis.

Answer: The reviewer’s concern about the expression of SLC7A11 in MCF-7 and MDA-MB-231 cells was addressed in comments on Fig. 6 (see below). We have now included 1 μ M Liproxstatin-1 in Fig. 5b, 5c, 5f, and 5i and observed that Liproxstatin-1 significantly reduced Erastin-induced cell death in MCF-7, MDA-MB-231 and 4T1 cells by cell count assay or MTT assay, indicating that Erastin-induced cell death in these cells was mediated by ferroptosis.

Fig. 5. b, Cell viability of MCF-7, MDA-MB-231 or 4T1 cells cultured with respective control CM \pm of 1 μ M Liproxstatin-1 or BCSC CM \pm 5 μ M Gallicyanine and treated with a graded concentration of Erastin. **c,** MCF-7, MDA-MB-231 or 4T1 cells were co-cultured with respective parental cells \pm 1 μ M Liproxstatin-1 or BCSCs \pm 5 μ M Gallicyanine and treated with 10 μ M (MCF-7), 5 μ M (MDA-MB-231) and 1 μ M (4T1) Erastin, respectively. Cell viability was determined by cell count assay. **f,** Cell viability of MCF-7, MDA-MB-231 or 4T1 cells were cultured with 100 ng/ml recombinant DKK1 or 1 μ M Liproxstatin-1 and treated with a graded concentration of Erastin for 48 hours. **i,** Cell viability of MDA-MB-231-shCONT or -shDKK1 cells \pm 1 μ M Liproxstatin-1 and treated with a graded concentration of Erastin for 48 hours.

Figure 5d. It should be stated what was the concentration of Erastin used, and what was the incubation time

Answer: In Fig. 5d, MCF-7, MDA-MB-231 and 4T1 cells were treated with 10 μ M, 5 μ M and 1 μ M for 48 hours, respectively. We have now included this information in the figure legend.

Figure 5i. It seems like Liproxstatin-1 only rescues the effect of DKK1 depletion in Erastin treated cells, but not the cell death caused by Erastin. How can this be explained?

Answer: We have repeated this experiment with newly purchased Liproxstatin-1. As shown below, we observed a more efficient rescue effect of Liproxstatin-1 in DKK1 depleted cells treated with Erastin. However, application of 1 μ M Liproxstatin-1 still could not completely abrogate the cell death induced by high concentrations of Erastin. This may be due to the relative ratio of the concentrations, in that 1 μ M Liproxstatin-1 is insufficient to rescue 10 μ M or 20 μ M Erastin-induced cell death.

Fig. 5i, Cell viability of MDA-MB-231-shCONT or -shDKK1 cells in the presence or absence of 1 μ M Liproxstatin-1 and treated with a graded concentration of Erastin for 48 hours.

l. 436: when 4T1 cells are pretreated for 48h with Erastin, according to Fig 5b, all cells should be dead, or did the authors use a sublethal concentration?

Answer: In Fig. 6k, 4T1 cells were actually pre-treated with 1 μ M Erastin for 48 hours and subsequently injected into the tail vein of BALB/c mice. The 1 μ M Erastin is a sublethal concentration and did not kill all the cells. Each group was injected with the same number of viable cells. The concentration of Erastin has now been included in the figure legend.

Figure 6. Many currently available antibodies against SLC7A11 are non-specific and unsuitable for determining SLC7A11 levels (Jiang et al Nat Rev Mol Cell Biol 2021). The authors use an antibody that detects a band size of 50 kD. Normally SLC7A11 should run at 35-40 kD. Using a validated antibody, it was shown that SLC7A11 is hardly expressed on MDA-MB-231 or MCF-7 cells (for reference check Doll et al. Nature 2019). Therefore, all data using the current antibody should be repeated using a suitable antibody.

Answer: The monomeric SLC7A11 contains 501 amino acids and has a predicted molecular weight of 55 kDa. However, it has a high percentage of hydrophobic amino acid residues which may promote the migration of the protein in SDS-PAGE. Endogenous monomeric SLC7A11 is expected to migrate at ~35 kDa and modified-SLC7A11 is expected to migrate at ~55 kDa⁴. The different observed weights of SLC7A11 may be due to differential modifications. The reported low expression of SLC7A11 in MCF-7 and MDA-MB-231 cells in the reference paper (Doll et al. Nature 2019) may be due to two reasons: 1. The high intensity signal of SLC7A11 in MDA-MB-436 and HS-578T cells in the same western blot exposure results in its relative underexposure in MCF-7 and MDA-MB-231 cells; 2. The predominant observed weight of SLC7A11 in MCF-7 and MDA-MB-231 cells is ~55 kDa, and the ~35 kDa band is hardly observed as reported by Doll et al, 2019.

The antibody used in this study was purchased from Proteintech (Catalog number: 26864-1-AP), the authenticity of which is validated by knockdown experiment in our data (Extended Data Fig. 7b) and in other published studies^{5,6}. To further address the reviewer's concern, we performed the immunoblot assessment of SLC7A11 using another antibody from Abcam (Catalog number: ab175186). Both molecular weights (~35 kDa⁷ and ~55 kDa^{8,9}) were previously observed using the Abcam antibody. We therefore performed immunoblot assessment of SLC7A11 in MCF-7, MDA-MB-231

and 4T1 cells by using this antibody, both ~55 KDa and ~35 KDa bands were observed in these cells (as shown below), but the intensity signal at ~55 KDa was higher than that at ~35 KDa, indicating that the dominant form of SLC7A11 in these cells may be the modified-SLC7A11 at ~55 KDa. All the immunoblot data for SLC7A11 has been repeated using the antibody from Abcam and consistent results were observed (Fig. 6b, 6c, 6f, 6g and Extended Data Fig. 7b). All data using SCL7A11 antibody (Proteintech, 26864-1-AP) previously has been replaced by using the new antibody from Abcam (Catalog number: ab175186).

The uncropped images of the immunoblot assessment of SLC7A11 in MCF-7, MDA-MB-231 and 4T1 cells using the antibody from Abcam (ab175186).

Galloyanine is a phenoxazine derivative; as phenoxazine is a privileged scaffold for ferroptosis inhibitors or radical-trapping antioxidants (RTA) (see Farmer et al JOC 2021), did the authors check whether Galloyanine has potential RTA activity, for example at high concentrations?

Answer: We thank the reviewer for the suggestion and have now measured the RTA activity of Galloyanine at concentrations of 2 μ M, 5 μ M and 10 μ M. As shown below, no significant RTA activity of Galloyanine was observed.

Co-oxidation of 1,4-dioxane (2.9 M) in 2:1 PhCl/DMSO initiated with 6 mM AIBN at 50 °C. PBD-BODIPY was used as the signal carrier for autoxidation, and the reaction progress was monitored at 587 nm.

References

1. Liu S, *et al.* Breast cancer stem cells transition between epithelial and mesenchymal states reflective of their normal counterparts. *Stem Cell Reports* **2**, 78–91 (2014).
2. Ubellacker JM, *et al.* Lymph protects metastasizing melanoma cells from ferroptosis. *Nature* **585**, 113–118 (2020).
3. Elgendy SM, Alyammahi SK, Alhamad DW, Abdin SM, Omar HA. Ferroptosis: An emerging approach for targeting cancer stem cells and drug resistance. *Crit Rev Oncol Hemat* **155**, (2020).
4. Shih AY, Erb H, Sun X, Toda S, Kalivas PW, Murphy TH. Cystine/glutamate exchange modulates glutathione supply for neuroprotection from oxidative stress and cell proliferation. *J Neurosci* **26**, 10514–10523 (2006).
5. Ni HW, *et al.* MiR-375 reduces the stemness of gastric cancer cells through triggering ferroptosis. *Stem Cell Res Ther* **12**, (2021).
6. Xu XT, *et al.* Targeting SLC7A11 specifically suppresses the progression of colorectal cancer stem cells via inducing ferroptosis. *Eur J Pharm Sci* **152**, (2020).
7. Guan ZH, Chen J, Li XL, Dong N. Tanshinone IIA induces ferroptosis in gastric cancer cells through p53-mediated SLC7A11 down. *Bioscience Rep* **40**, (2020).
8. Saito T, *et al.* p62/Sqstm1 promotes malignancy of HCV-positive hepatocellular carcinoma through Nrf2-dependent metabolic reprogramming. *Nature Communications* **7**, (2016).
9. Fan BY, *et al.* Liproxstatin-1 is an effective inhibitor of oligodendrocyte ferroptosis induced by inhibition of glutathione peroxidase 4. *Neural Regen Res* **16**, 561–566 (2021).

Reviewers' Comments:

Reviewer #1:

Remarks to the Author:

The authors have satisfactorily addressed my concerns.

The English requires further editing.

Reviewer #2:

Remarks to the Author:

In the revised manuscript by Wu et al. "Cancer Stem Cell Regulated Phenotypic Plasticity Protects Metastasized Cancer Cells from Ferroptosis" the authors attempted to address all concerns.

However few questions or comments remain:

- Figure 5: The authors have now included Liproxstatin-1 as a control. Yet, the authors or readers should be aware that erastin can cause unspecific toxicities at high concentrations, which are unrelated to ferroptosis. This is also reflected here, as complete rescue of erastin-induced cell death with Liproxstatin-1 was not possible when using high concentrations of erastin.

- Figure 5i: I do not agree with the authors interpretation of the data:

"However, application of 1 μ M Liproxstatin-1 still could not completely abrogate the cell death induced by high concentrations of erastin. This may be due to the relative ratio of the concentrations, in that 1 μ M Liproxstatin-1 is insufficient to rescue 10 μ M or 20 μ M erastin-induced cell death."

Here, I assume that erastin causes unspecific toxicities at a concentration of 10 μ M or 20 μ M, that can only partially be rescued by a ferroptosis inhibitor such as Liproxstatin-1.

-Figure 6: Thank you for providing further data on alternative SLC7A11 antibodies. However, as previously pointed out I do recommend the use of a specific anti-SLC7A11 antibody as published by Doll et al. Nature 2019. This antibody is also commercially available (Sigma/Merck Anti-xCT, clone 3A12, Cat. No. MABC1637). Unlike suggested by the authors, Doll et al. shows pictures of whole blots in the supplement. While there are some unspecific bands at 55 kDa in certain cell lines, the prominent and specific band is shown at \sim 35-40 kDa.

REVIEWER COMMENTS

Reviewer #1 (Remarks to the Author):

The authors have satisfactorily addressed my concerns.

The English requires further editing.

Answer: We have revised the manuscript carefully with optimized editing of the language.

Reviewer #2 (Remarks to the Author):

In the revised manuscript by Wu et al. "Cancer Stem Cell Regulated Phenotypic Plasticity Protects Metastasized Cancer Cells from Ferroptosis" the authors attempted to address all concerns. However few questions or comments remain:

- Figure 5: The authors have now included Liproxstatin-1 as a control. Yet, the authors or readers should be aware that erastin can cause unspecific toxicities at high concentrations, which are unrelated to ferroptosis. This is also reflected here, as complete rescue of erastin-induced cell death with Liproxstatin-1 was not possible when using high concentrations of erastin.

Answer: We appreciate the valuable comments and agree with the reviewer that Erastin can cause nonspecific toxicities at high concentrations. However, although the possible toxicities of Erastin at high concentrations should be noted, our data showed that BCSC-secreted DKK1 reduced Erastin-induced cell death at both high and low concentrations of Erastin, supporting the notion that BCSC-secreted DKK1 protected cells from ferroptosis. This observation was validated by another ferroptosis inducer RSL3 (Extended Data Fig. 6b and 6d-6e). Additionally, BCSC-secreted DKK1 also reduced Erastin-induced lipid ROS accumulation (Fig. 5d), further supporting this conclusion.

In order to avoid possible misinterpretation regarding the non-specific toxicity of high concentrations of Erastin, we have now modified the description of the correlation between Erastin and ferroptosis in the revised manuscript.

The description in the revised manuscript now states:

Application of liproxstatin-1, a ferroptosis antagonist, partially reduced Erastin-induced cell death at high concentrations of Erastin (Fig. 5b). Although the non-specific toxicities of Erastin at high concentrations should also be noted, all cells cultured with BCSC CM developed persistent resistance to Erastin-induced ferroptosis at both high and low concentrations of Erastin, whereas DKK1 inhibitor addition to BCSC CM significantly ameliorated this resistance (Fig. 5b).

- Figure 5i: I do not agree with the authors interpretation of the data:

"However, application of 1 μ M Liproxstatin-1 still could not completely abrogate the cell death induced by high concentrations of erastin. This may be due to the relative

ratio of the concentrations, in that 1 μM Liproxstatin-1 is insufficient to rescue 10 μM or 20 μM erastin-induced cell death.”

Here, I assume that erastin causes unspecific toxicities at a concentration of 10 μM or 20 μM , that can only partially be rescued by a ferroptosis inhibitor such as Liproxstatin-1.

Answer: Similar to the answer above, we have now modified the description of these data in the revised manuscript.

-Figure 6: Thank you for providing further data on alternative SLC7A11 antibodies. However, as previously pointed out I do recommend the use of a specific anti-SLC7A11 antibody as published by Doll et al. Nature 2019. This antibody is also commercially available (Sigma/Merck Anti-xCT, clone 3A12, Cat. No. MABC1637). Unlike suggested by the authors, Doll et al. shows pictures of whole blots in the supplement. While there are some unspecific bands at 55 kDa in certain cell lines, the prominent and specific band is shown at ~ 35-40 kDa.

Answer: We thank the reviewer for the kind and important comments. To assist in resolving this issue, we first consulted the local Chinese supplier for Sigma/Merck products and unfortunately the anti-xCT antibody (Sigma/Merck, Catalog number: MABC1637) is not available in China (The communication letter with the company is attached).

It is possible that different anti-SLC7A11 antibodies may target different epitopes. The monoclonal antibody to SLC7A11 used in the referenced paper (Doll et al. Nature 2019) may selectively recognize the modified-SLC7A11 at ~35 kDa, and the recombinant antibody of SLC7A11 (Abcam, Catalog number: ab175186) may preferentially recognize the modified-SLC7A11 at ~55 kDa, which may contribute to different abundances of the observed bands.

We have now also performed the immunoblot assessment of SLC7A11 using another antibody from Cell Signaling Technology (CST, Catalog number: #12691). Both molecular weights at ~35 kDa and ~55 kDa were previously observed using this CST antibody (Weimin Wang et al. Nature 2019, referenced in the uncropped gel data of Fig. 2g in Supplementary Figures). As shown below, a predominant band at ~55 kDa was observed in MCF-7 and MDA-MB-231 cells by using this anti-SLC7A11 antibody from CST. The shRNA targeting SLC7A11 could specifically diminish the ~55 kDa band recognized by all the three anti-SLC7A11 antibodies that were used (Proteintech, Abcam and CST), suggesting the specificity of these anti-SLC7A11 antibodies predominantly at ~55 kDa. Hence, this data indicates the expression of modified-SLC7A11 at ~55 kDa in MCF-7 and MDA-MB-231 cells.

Figure: The uncropped images of the immunoblot assessment of SLC7A11 in MCF-7 and MDA-MB-231 using the antibody from CST (Catalog number: #12691).

To further validate whether SLC7A11 is expressed in MCF-7 and MDA-MB-231 cells, we analyzed the RNA sequencing data of MCF-7 (the RNA sequencing data used in this study) and MDA-MB-231 cells (unpublished RNA sequencing data performed in our lab). We observed that the mRNA levels of SLC7A11 in MCF-7 and MDA-MB-231 cells were close to the median value of the overall genes, suggesting a considerable expression of SLC7A11 in these cells.

Figure: The expression pattern of protein-coding genes in MCF-7 and MDA-MB-231 cells.

Communication letters

From: wumm2012@mail.ustc.edu.cn <wumm2012@mail.ustc.edu.cn>

Sent: Wednesday, December 8, 2021 11:45 AM

To: China Orders <ordercn@merckgroup.com>

Subject: Inquiry about the anti-xCT antibody

Dear Customer Service Team of the Merck Co.,

Thank you for replying to my inquiry about the anti-xCT antibody (Sigma/Merck, Catalog number: MABC1637) over the phone. My understanding is that this antibody is not for sale in China. In order to avoid any misunderstanding, I would like to double confirm with you. Thanks for your assistance and looking forward to your reply.

Sincerely,

Mingming Wu

Reply:

Dear customers,

As for the anti-XCT antibody (Sigma/Merck, catalog number: MABC1637) you consulted on the phone, it cannot be sold in China at present.

Thanks and best regards,

Shushu Lin

Contact Center – Commercial Care Team

Life Science | Research & Applied Solutions | Customer Excellence

Merck

17/F, No.3, Building C, The New Bund World Trade Center | Shanghai 200126 | China

Contact Center: 400 620 3333 | 800 819 3336

Ordering: ordercn@merckgroup.com

Technical Service: tscn@merckgroup.com

Reviewers' Comments:

Reviewer #2:

Remarks to the Author:

The authors have addressed all the remaining comments.

REVIEWERS' COMMENTS

Reviewer #2 (Remarks to the Author):

The authors have addressed all the remaining comments.

Answer: We acknowledge the reviewer for the positive and valuable comments, which have greatly improved this study.